# Species clustering, climate effects, and introduced species in 5 million city trees across 63 US cities

Dakota E McCoy[1,2,3,4†], Benjamin Goulet-Scott[1,5*†], Weilin Meng[6], Bulent Furkan Atahan[7], Hana Kiros[1], Misako Nishino[8], John Kartesz[8]

[1]Department of Organismic and Evolutionary Biology, Harvard University, Cambridge, United States; [2]Department of Materials Science and Engineering, Stanford University, Stanford, United States; [3]Hopkins Marine Station, Stanford University, Pacific Grove, United States; [4]Department of Biology, Duke University, Durham, United States; [5]Harvard Forest, Harvard University, Petersham, United States; [6]Independent Researcher, Boston, United States; [7]Department of Biology and Biotechnology, Worcester Polytechnic Institute, Worcester, United States; [8]The Biota of North America Program (BONAP), Chapel Hill, United States

*For correspondence:
bgoulet@g.harvard.edu

[†]These authors contributed equally to this work

Competing interest: The authors declare that no competing interests exist.

**Abstract** Sustainable cities depend on urban forests. City trees—pillars of urban forests—improve our health, clean the air, store $CO_2$, and cool local temperatures. Comparatively less is known about city tree communities as ecosystems, particularly regarding spatial composition, species diversity, tree health, and the abundance of introduced species. Here, we assembled and standardized a new dataset of $N$ = 5,660,237 trees from 63 of the largest US cities with detailed information on location, health, species, and whether a species is introduced or naturally occurring (i.e., "native"). We further designed new tools to analyze spatial clustering and the abundance of introduced species. We show that trees significantly cluster by species in 98% of cities, potentially increasing pest vulnerability (even in species-diverse cities). Further, introduced species significantly homogenize tree communities across cities, while naturally occurring trees (i.e., "native" trees) comprise 0.51–87.4% (median = 45.6%) of city tree populations. Introduced species are more common in drier cities, and climate also shapes tree species diversity across urban forests. Parks have greater tree species diversity than urban settings. Compared to past work which focused on canopy cover and species richness, we show the importance of analyzing spatial composition and introduced species in urban ecosystems (and we develop new tools and datasets to do so). Future work could analyze city trees alongside sociodemographic variables or bird, insect, and plant diversity (e.g., from citizen-science initiatives). With these tools, we may evaluate existing city trees in new, nuanced ways and design future plantings to maximize resistance to pests and climate change. We depend on city trees.

## Editor's evaluation

This paper will be of interest to urban foresters, ecologists, and planners. It provides a large new dataset of city tree communities across US cities, which may ignite new studies on city biodiversity and ecosystem services. It contains clear descriptions about the data processing and structures, and the potential uses of the data.

**eLife digest** Trees in towns and cities provide critical services to humans, animals and other living things. They help prevent climate change by capturing and storing carbon dioxide; they provide food and shelter to other species, they scrub the air of microscopic pollutants, cool local temperatures, and improve the mental and physical health of those who have access to them.

In general, naturally occurring (so called native) plant species support richer local ecosystems – such as bird and butterfly communities – than plants that have been introduced from other areas. However, relatively little is known about which species of trees are found in towns and cities or how these species are distributed.

Here, McCoy, Goulet-Scott et al. assembled a dataset of 5.6 million city trees from 63 cities in the United States. This dataset contained rich data on the exact location, species, and health of individual city trees – including park trees, those in urban forests, and trees that line city streets.

In nearly all of the cities, the same tree species were found clustered next to each other, even in cities that had many different species of tree overall. This tendency of tree species to flock together may make these communities more vulnerable to disease and pest outbreaks. Trees in more developed environments, like those that line streets, were much less species diverse than trees spread across parks.

Cities with wetter, cooler climates tended to have higher percentages of native tree species compared to cities with drier, hotter climates. Younger cities also had a greater percentage of native tree species than older cities, which may reflect increased awareness of the importance of native tree species among urban planners in more recent years. The cities that had planted non-native tree species tended to select the same species, which contributed to tree communities in different cities looking more alike.

McCoy, Goulet-Scott et al. provide easy-to-use tools academics and urban foresters can use to assess how diverse tree communities in individual cities are. This work may help local decision-makers to select and plant trees that build resilience against climate change, pest and disease outbreaks, and maximize the health benefits trees provide all city dwellers.

## Introduction

Cities are ecosystems. Humans (*Willis and Petrokofsky, 2017*) and other animals (*Berthon et al., 2021*) depend on urban forests, which are the woody and associated vegetation in and around dense human settlements (*Konijnendijk et al., 2006*). City tree communities, an essential component of urban forests, improve our cities in many ways. City trees boost mental and physical health (*Hartig and Kahn, 2016*), capture and store carbon dioxide (*Rowntree and Nowak, 1991*), scrub toxic particulate matter from the air (*Nowak et al., 2014*), and cool local temperatures by about 0.83°C for every 10% increase in forest cover (*Kong et al., 2014*). The financial benefits of having a tree-rich city—rather than a concrete jungle—are huge and well documented (*McPherson et al., 2016*). Tree inventories provide a wealth of useful data (*Cowett and Bassuk, 2014*; *Cowett and Bassuk, 2020*; *Galle et al., 2021*; *Kendal et al., 2014*; *Love et al., 2022*; *McPherson et al., 2016*; *Ossola et al., 2020*; *Richards, 1983*; *Steenberg, 2018*). Many studies underscore the importance of city plant life to humans, but comparatively fewer evaluate urban forests as potentially biodiverse ecosystems (*Alvey, 2006*). Through this ecological lens, it is important to understand species diversity (*Behm, 2020*), nativity status (*Tallamy, 2004*), and spatial arrangements of city trees (*Roman et al., 2018*). In particular, we wanted to know whether local climatic conditions are associated with the species diversity of city tree communities, how species diversity was distributed in space within cities, and whether introduced tree species contribute to biotic homogenization among urban ecosystems.

Here, we assembled a dataset of $N$ = 5,660,237 individual trees from 63 US cities (*Figure 1—source data 1*; https://doi.org/10.5061/dryad.2jm63xsrf) with data on species, exact location, nativity status (naturally occurring vs. introduced), and standardized health (tree condition). We also developed tools to analyze the diversity, spatial structure, abundance of naturally occurring versus introduced trees, and overall condition of city tree communities. We demonstrate that these new tools provide a richer picture of city trees than relying on canopy cover and species count alone. For example, it is now possible for researchers to assess the spatial arrangement of trees by species (taking into

consideration the underlying spatial structure of city streets)—a metric which, we show, is not dependent on tree species diversity and which may indicate vulnerability to pests such as Dutch Elm disease (*Laćan and McBride, 2008*). Likewise, we show that the abundance of introduced trees varies greatly, even among cities with a high diversity of tree species; abundance of naturally occurring trees (i.e., "native" trees) is a useful proxy for an environment's capacity to support diverse communities of birds, butterflies, and other animals (*Burghardt et al., 2009*; *Burghardt et al., 2010*; *Tallamy, 2004*).

Taken together, we make available a large new dataset of city trees, user-friendly tools to better analyze the ecosystem structure of city tree communities, and proof-of-concept analyses to demonstrate potential uses of the data. Through these technical and practical advances, we help to enable the design of rich, heterogenous ecosystems built around city trees.

## Results and discussion
### A new dataset of more than 5 million city trees
First, we assembled and standardized a large dataset of $N$ = 5,660,237 city trees to enable the analysis of urban forests' ecosystem structure. We acquired tree inventories from 63 of the largest 150 US cities (those which had conducted inventories) and developed a standardization pipeline in *R* and *Python* (*Source code 1*). Each inventory was produced using different, city-specific methods: for example, some cities only reported a tree's common name; some reported an address but no coordinates; some reported tree size in feet, some in meters; some scored tree health from 1 to 5 while others rated trees as 'good' or 'poor'; very few cities reported whether each tree was an introduced species; etc. Therefore, we inspected metadata for all cities and communicated with urban officials to standardize column names, standardize metrics of tree health, and convert all units to metric (*Supplementary file 1*; *Source code 1*; 'Materials and methods'). We converted all common names to scientific and manually corrected misspellings in all species names (see *Source code 1*, and 'Materials and methods', for full details). We manually coded all tree locations as being in a green space or in an urban environment to enable comparisons between location types. Finally, we referenced data from the Biota of North America Project on nativity status to classify each tree as naturally occurring or introduced. The resulting dataset (*Figure 1*, *Figure 1—source data 1*) comprised 63 city datasheets each with 28 standardized columns (*Supplementary file 1*).

### New tools for—and preliminary analyses of—species diversity, spatial structure, introduced species, tree health, and climate effects
Typically, researchers analyze city tree communities through species richness (as a measure of diversity) and percent canopy cover. Our large, fine-grained dataset allows for analysis of (1) effective species counts (a robust measure of diversity defined as the exponent of the Shannon–Weiner index; *Equation 1*), (2) spatial structure of city tree communities, (3) abundance of introduced versus naturally occurring trees, (4) climate drivers of species diversity and naturally occurring tree abundance, and (5) how city tree diversity correlates with fine-grained data on socioeconomics, demographics, the physical environment, and other forms of species diversity (e.g., birds and insects).

We found that city tree communities are moderately biodiverse, particularly in parks (*Figure 2*), but are significantly clustered by individual species (*Figure 3*). City tree communities varied in number of species represented (min = 16, median = 137, max = 528; *Figure 1—source data 2*) and in a robust, naturalistic measure of species diversity known as effective species count (min = 6 to max = 93 with a median = 26; *Figure 2A*). Tree communities located in parks were significantly more diverse than trees located in developed environments (e.g., along streets), controlling for population size (*Figure 2B*, *Figure 2—figure supplement 1*). For all analyses, when comparing diversity measures across different size scales, we applied rarefaction and extrapolation techniques using the R package iNext (see 'Materials and methods'; *Chao et al., 2015*; *Chao et al., 2014*; *Chao and Jost, 2012*; *Hsieh et al., 2016*) and performed sensitivity analyses excluding low-coverage cities.

Another commonly used species diversity metric is maximum abundance: relative abundance or frequency of the most abundant species. Many foresters follow Santamour's 10/20/30 rule, that the relative abundance of the most common species in a city should be less than 10%, the most common genus less than 20%, and the most common family less than 30% (*Santamour, 2004*). Here, the relative abundance of the most common species correlated significantly with effective species number,

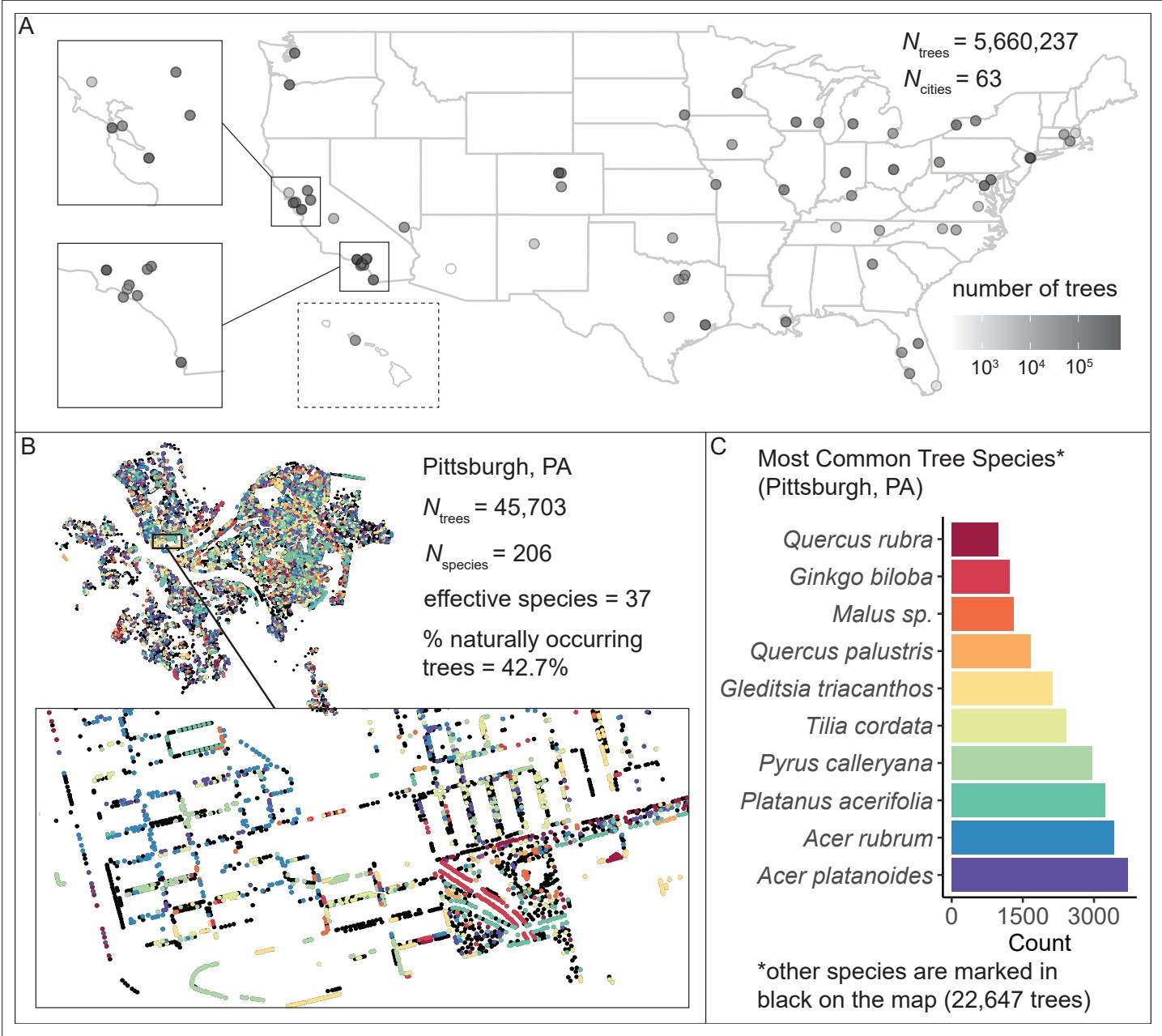

**Figure 1.** We assembled and standardized a dataset of *N* = 5,660,237 street trees from publicly available street tree inventories across 63 cities in the USA. (**A**) The number of trees recorded per city varied from 214 (Phoenix, AZ) to 720,140 (Los Angeles, CA) with a median of 45,148. (**B**) Sample plot of Pittsburgh, PA with trees colored by species type (inset: zoomed-in view of trees lining streets and parks). We include statistics for total number of trees $N_{trees}$ = 45,703; total number of species $N_{species}$ = 206; effective species count = 36 (a measure of diversity that incorporates both richness [number of species] and evenness [distribution of those species]; see *Equation 1*); and percent naturally occurring (rather than introduced) trees = 42.7%. (**C**) Counts of the 10 most common species inventoried in Pittsburgh; not shown are 22,647 trees belonging to other species (black points in (**B**)). The dataset includes information on species, exact location, whether a tree is introduced or naturally occurring, tree height, tree diameter, location type (green space or urban setting), tree health/condition, and more (*Figure 1—source data 1*). Source data are *Figure 1—source data 1* and *Figure 1—source data 2*; source code is *Source code 2*.

The online version of this article includes the following source data for figure 1:

**Source data 1.** City_Trees_Data_63_Files.zip.

**Source data 2.** Tree_Data_Summary_By_City.csv.

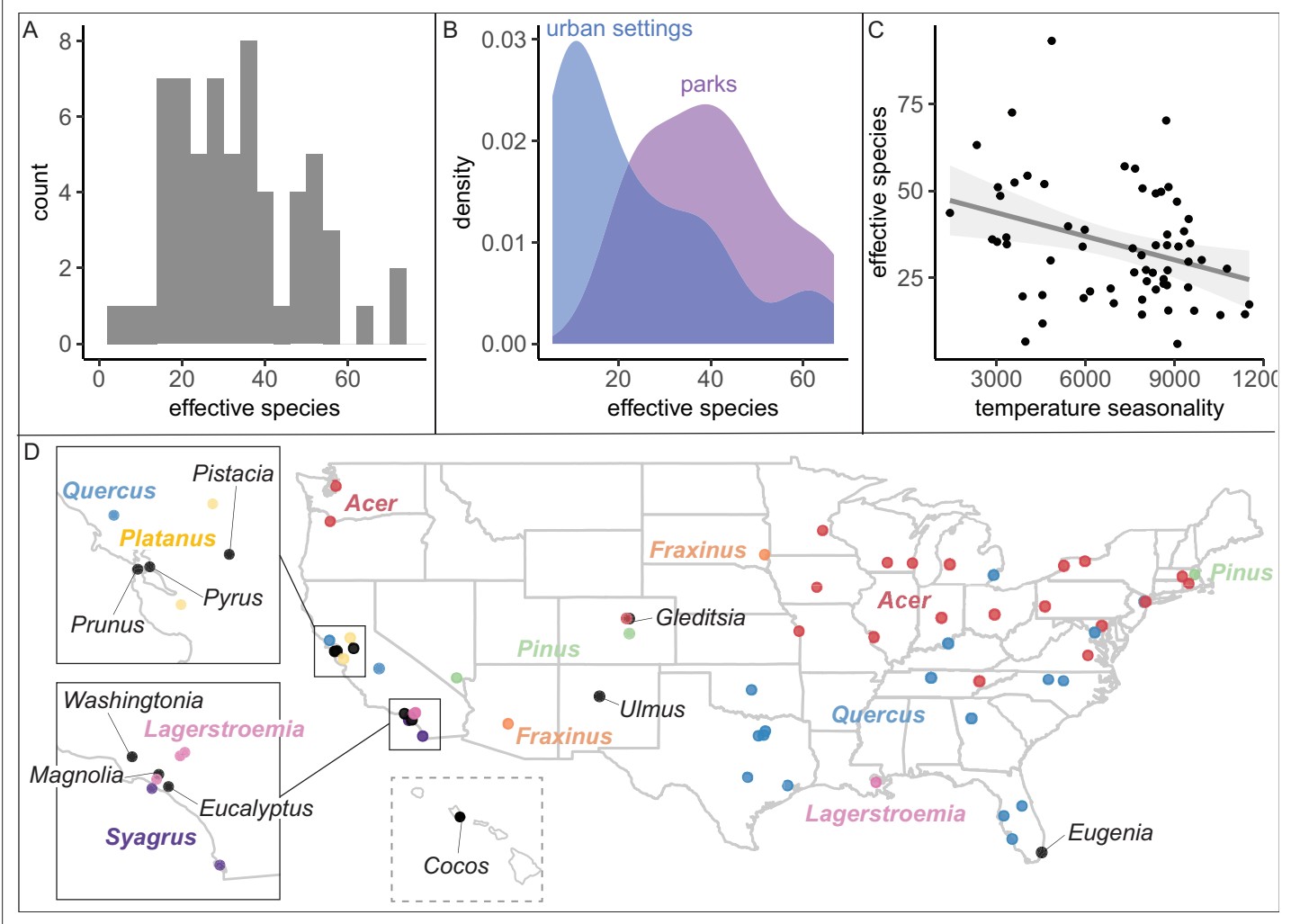

**Figure 2.** City tree communities are diverse and shaped by climate, although certain genera dominate. (**A**) Effective species count, a measure of species diversity, ranged across cities from min = 6 to max = 93 with a median = 26. We use Shannon's effective species count (***Equation 1***), a more nuanced metric than abundance-based metrics (see ***Figure 2—figure supplement 2***). (**B**) Trees in parks were significantly more diverse than trees in urban settings such as along streets (two-sample paired $t$-test comparing effective species numbers; $t = 7$, $p < 0.0005$, 95% confidence interval [CI] = [11.8, 22.9], mean diff. = 17, degrees of freedom = 10.4). To account for differences in population size and sampling effort between areas, we calculated effective species number for a given population size (the smaller of the two options, park and urban, for each city) using rarefaction approaches in the R package *iNext*. Results were also significant for (1) raw effective species number and (2) asymptotic estimate of effective species number. See ***Figure 2— figure supplement 1*** for sample sizes. (**C**) Environmental factors were significantly correlated with effective species count, across six sensitivity conditions controlling for sampling effort, population size, and more (***Supplementary file 2***). Most sociocultural variables were not significant, but cities designated as 'Tree City USA' were significantly more likely to have higher effective species numbers than those without that designation (for three of our six sensitivity analyses). Here, we plot the negative relationship between tree species diversity (effective species count controlling for population size) and temperature seasonality (captured through environmental PC1; see ***Supplementary file 5***). To allow for comparison across cities with different sizes and sampling efforts, we plot the calculated effective species number for a population = 37,000 trees, the rounded median population size (using rarefaction and extrapolation in R package *iNext*). Results were also significant for (1) raw effective species number, (2) asymptotic estimate of effective species number, and when excluding cities with low sample size or sample coverage (***Supplementary file 2***). (**D**) The most abundant genus in each city is labeled here; see the most common species by city in ***Figure 2—figure supplement 3***. Supporting figures for this figure include ***Figure 2— figure supplement 1***, ***Figure 2—figure supplement 2***, and ***Figure 2—figure supplement 3***; ***Supplementary file 2*** and ***Supplementary file 5*** are supporting tables. Source data are ***Figure 1—source data 1***, ***Figure 1—source data 2***, and ***Figure 2—source data 1***; source code is ***Source code 2***; and an associated tool to calculate effective species is ***Source code 3***.

The online version of this article includes the following source data and figure supplement(s) for figure 2:

**Source data 1.** Rarefaction_Plots.zip.

**Figure supplement 1.** Tree communities in parks were significantly more biodiverse than those in urban settings but did not differ significantly in percent of trees that were naturally occurring (rather than introduced).

*Figure 2 continued on next page*

*Figure 2 continued*

**Figure supplement 2.** Effective species is a more nuanced metric of species diversity than the metric of the relative abundance of the most common species or genus.

**Figure supplement 3.** The most common tree species in each city is labeled.

but cities below the 10% max abundance threshold vary from 33 to 93 effective species (*Figure 2—figure supplement 2*). Therefore, Santamour's rule may be a necessary but not a sufficient guideline, so we developed an Excel resource to calculate effective species number from a list of (1) species counts or (2) all trees (*Source code 3*).

Because our dataset spans many different environmental conditions, we could assess the extent to which climate has impacted the ecosystem structure of city trees. We summarized the climate of each city with a principal components analysis (PCA) of 19 bioclimatic variables from the World-Clim (*Fick and Hijmans, 2017*) database (*Supplementary file 5*). Across the USA, climate—but not sociocultural factors—correlated with city tree species diversity (*Figure 2C*, *Supplementary file 2*). Specifically, controlling for sample size and coverage, temperature and rainfall significantly correlate with effective species count, aligning with previous analyses of city trees, *Kendal et al., 2014* and global distributions of plants, *Woodward and Williams, 1987*. Maples (*Acer*) and Oaks (*Quercus*) dominated city tree genera across the country (*Figure 2D*), while the most common species were *Acer platanoides* (Norway Maple), *Fraxinus pennsylvanica* (Green Ash), *Lagerstroemia indica* (crape myrtle), and *Platanus acerifolia* (London plane); see *Figure 2—figure supplement 3*.

We next investigated the spatial arrangement of species diversity in city tree communities. Species-diverse, rather than species-poor, city tree communities offer many well-documented benefits. Species-diverse forests are more effective in resisting diseases (*Laćan and McBride, 2008*), are more resilient in the face of climate change (*Roloff et al., 2009*) and confer greater mental health benefits (*Fuller et al., 2007*). Compared to species diversity, the spatial arrangement of trees is less well

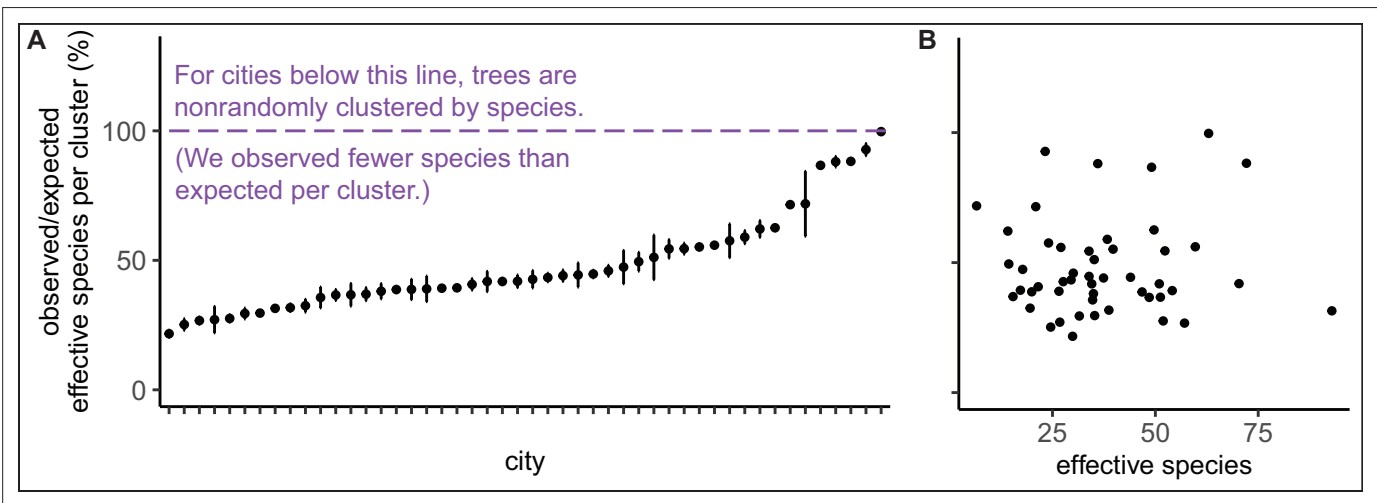

**Figure 3.** Trees are spatially clustered by species in nearly all cities, even in cities with high species diversity. (**A**) In 47 of 48 cities, trees are non-randomly clustered by individual species (with significantly fewer effective species per spatial cluster than expected, i.e., values <100%). Plotted points represent median values and 95% confidence intervals (observed/expected effective species counts) for all clusters in a city (see $N_{clusters}$ per city and full statistics in *Figure 3—source data 1*). We excluded one city, Greensboro, from the analysis due to insufficient sample size (10 clusters). (**B**) The degree of spatial clustering in a city was not correlated with effective species number, a measure of tree diversity (*Figure 3—figure supplement 1*). To control for different sizes and sampling efforts across cities, here we plot the calculated effective species number for a given population = 37,000 trees (using rarefaction and extrapolation in R package *iNext*). $N_{cities}$ = 48. *Figure 3—figure supplement 1* is a supporting figure for this figure. Source data are *Figure 1—source data 1* and *Figure 3—source data 1*; source code is *Source code 2*.

The online version of this article includes the following source data and figure supplement(s) for figure 3:

**Source data 1.** Clustering_Results.csv.

**Figure supplement 1.** The degree to which a city has trees clustered by species does not correlate with abundance-based measures of species diversity.

understood, even though clusters of same species of trees may be more susceptible to pest outbreaks (*Greene and Millward, 2016*; *Raupp et al., 2006*).

We found that city trees were non-randomly clustered by individual species in 47 of 48 cities (*Figure 3A*). Additionally, a city's clustering score was not significantly correlated with species diversity metrics and is therefore a separate metric of interest (*Figure 3B*, *Figure 3—figure supplement 1*). City tree communities with well-mixed arrangements of trees may be more resistant to species-specific diseases and blights, as in the case of the Emerald Ash Borer *Agrilus planipennis* (*Greene and Millward, 2016*). Clustering by species is not necessarily a negative, nor indeed should we necessarily expect trees to be randomly arranged (see suggestions for further research in 'Future Analyses' section). Here, we take a first step toward making spatial clustering a metric of interest in city tree planning.

As city officials consider which trees to plant where, weighing many factors such as appearance and hardiness (*Conway and Vander Vecht, 2015*), we suggest they consider a simple metric of species clustering. To calculate clustering metrics, readers familiar with Python and R can use the code in *Source code 2*; others should contact the authors (a web resource is currently under development).

Our new dataset allows researchers and urban foresters to consider the utility of naturally occurring versus introduced trees (i.e., "native" vs. "non-native" trees). Whether or not a city decides to plant naturally occurring species rather than introduced species is a growing topic of interest (along with whether nativity status matters, and how to define "native" or "naturally occurring", *Berthon et al., 2021*; *Gould, 1998*; *Sjöman et al., 2016*). We classify plants as "naturally occurring" if they occur in a particular region without direct or indirect recent human intervention. This definition does not account for the substantial effects of Indigenous peoples on plant communities before European contact, nor does this paper address the flaws with a "native-or-not" ecological approach (see discussion of an alternative Indigenous ecology in *Grenz, 2020*; *McKay and Grenz, 2021*).

Here, we found that the percent of trees that were naturally occurring (i.e., "native") varied across cities from 0.51% to 87.4% with a median of 45.6% (*Figure 4*). Wetter, cooler climates correlated with significantly higher percentages of naturally occurring trees (*Figure 4A, B*). However, it is important to note a strong east-to-west gradient, by which more introduced trees were present in western states (*Figure 4A*). Thus, some social factor may have influenced the planting of introduced trees (*Roman et al., 2018*; *Steenberg, 2018*). However, after accounting for climate, younger cities had a higher percentage of naturally occurring trees (*Supplementary file 3*); perhaps urban forestry practitioners have been more likely to consider nativity status in recent years. The observed east-to-west gradient deserves further research attention.

In general, naturally occurring ("native") plant species support richer local ecosystems (e.g., more diverse and numerous bird and butterfly communities, *Burghardt et al., 2009*; *Burghardt et al., 2010*). Among introduced plants, those with naturally occurring congeners support more and more diverse Lepidopteran species than those without (*Burghardt et al., 2010*). Many cities with relatively low populations of naturally occurring trees nonetheless had many introduced trees with a naturally occurring congener (bottom right quadrant, *Supplementary file 5B*)—and therefore likely provide moderate insect habitat. Diversity of naturally occurring trees is significantly correlated with overall tree community diversity (*Figure 4—figure supplement 1*). Nativity status is a useful proxy for ecological value (although it is not, alone, a deciding factor, *Berthon et al., 2021*), so we developed an Excel tool to report nativity status as 'introduced' or 'naturally occurring' based on a user's list of species for a given city or state (*Source code 4*). Original BONAP data on all native taxa for each US state are available in *Figure 4—source data 1*.

Urban foresters typically aim to select tree species which will be healthy in their city environment. Our dataset provides standardized metrics of tree health across many cities, allowing analyses of what tree- or location-specific factors correlate with health in city trees. Our preliminary analyses suggest that whether or not a tree was an introduced species had no clear impact on tree health (*Supplementary file 4*). Trees are generally healthier when they are smaller and/or in an urban setting rather than in parks (*Supplementary file 4*), possibly because city arborists quickly remove unhealthy trees in densely populated areas where they pose a fall risk. Further work is needed on within-species trends.

Are city tree communities more similar to each other than we would expect based on geography and climate? Indeed, we found that introduced tree species drive similar species compositions between cities (*Figure 4C*), reflecting the phenomenon of 'biotic homogenization' (*McKinney and*

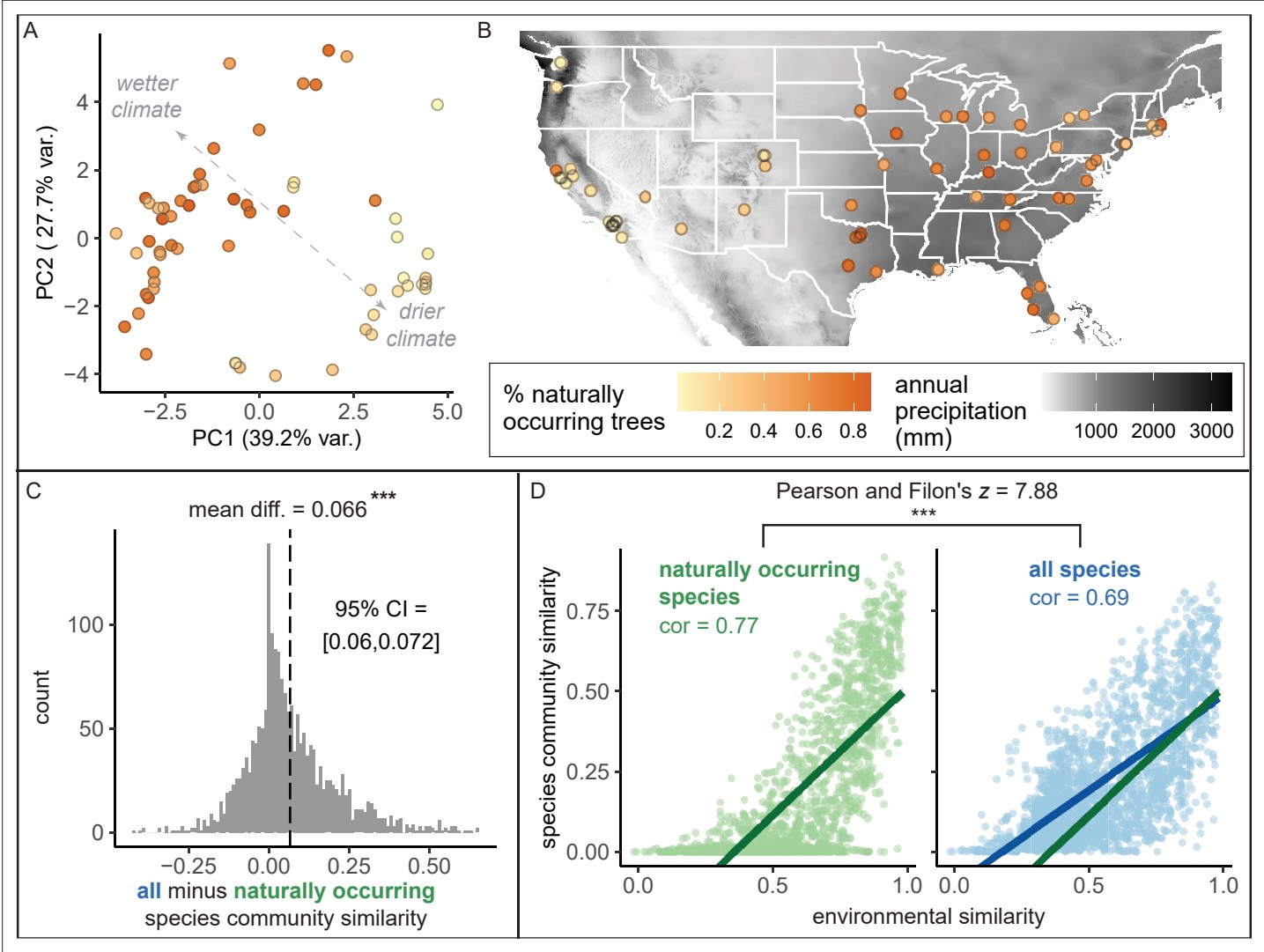

**Figure 4.** Environment strongly influences the percentage of naturally occurring trees, while introduced trees make species compositions more similar between cities. (**A**) Cities in wetter, cooler climates—and younger cities—had significantly higher percentages of naturally occurring (rather than introduced) trees (beta regression; AIC = −58.4, pseudo-$R^2$ = 0.64, log likelihood = 35.2; statistics in **Supplementary file 3**). Indeed, we found that wetter, cooler climates significantly predicted higher percentages of naturally occurring trees across four sensitivity tests: excluding outliers ($N_{cities}$ = 61); including cities with >10,000 trees ($N_{cities}$ = 49); including cities with >50% spatial coverage ($N_{cities}$ = 28); and including cities with high sample coverage ($N_{cities}$ = 56). See **Supplementary file 3**. Here, we plot a principal component analysis of the Bioclim variables (**Figure 4—source data 2**), colored by percent naturally occurring trees. Each point represents one city. Bioclim variables relating to precipitation (such as annual precipitation) are negatively correlated with PC1 and positively correlated with PC2 (see complete loadings in **Supplementary file 5**). (**B**) The percent of naturally occurring trees is plotted against annual precipitation in mm (black and white background). (**C**) Among city pairs ($N_{comparisons}$ = 1953), overall species communities are significantly more similar to one another than their naturally occurring species communities alone (paired $t$-test, $t$ = 20.4, p < 0.0005, 95% confidence interval [CI] = [0.060, 0.072], mean difference = 0.066, degrees of freedom = 1,952; result upheld by non-parametric Wilcoxon signed-rank test). We calculated chi-square similarity scores for each pair of cities under two conditions; first, we included all trees ('all'), then we included only naturally occurring trees ('naturally occurring'), and reported the difference between the two similarity scores. We controlled for different population sizes and sampling efforts by randomly subsampling the larger city in the pairwise comparison 50 times and calculating the median chi-squared similarity score from those 50 repetitions. (**D**) Among city pairs, environment is significantly more strongly related to naturally occurring species than introduced species. We compared chi-square similarity scores between species communities (left: naturally occurring only; right: all) against environmental similarity scores (one minus the normalized euclidean distance in our principal components analysis [PCA]). Left panel, green, naturally occurring species only: Pearson's product-moment correlation, cor = 0.77, 95% CI = [ 0.75, 0.78], $t$ = 52.7, p < 0.0005, degrees of freedom = 1,952. Right panel, blue, all species: Pearson's product-moment correlation; cor = 0.69, 95% CI = [0.67, 0.71], $t$ = 42.0, p < 0.0005, degrees of freedom = 1,952. In the right panel, the green line is the same as in the left panel to enable comparisons. **Figure 4—figure supplement 1** is a supporting figure for this figure, and **Supplementary file 3** and **Supplementary file 5** are supporting tables for this figure. Source data are **Figure 1—source data 1**, **Figure 1—source data 2**, **Figure 4—source**

*Figure 4 continued on next page*

*Figure 4 continued*

*data 1*, *Figure 4—source data 2*, and *Figure 4—source data 3*; source code is *Source code 2*; and an associated tool to label each species in a list of treespecies as 'naturally occurring' or 'introduced' is *Source code 4*. Significance level *** indicates p<0.0005.

The online version of this article includes the following source data and figure supplement(s) for figure 4:

**Source data 1.** Native_Taxa_By_State_BONAP.csv.

**Source data 2.** Environmental_PCA.xlsx.

**Source data 3.** Spatial_Coverage_Analysis.zip.

**Figure supplement 1.** The percent of trees that are naturally occurring rather than introduced is not the only important variable when considering nativity status; here, we plot the species diversity of naturally occurring trees and percent of introduced trees that are closely related to naturally occurring trees (have a naturally occurring congener).

---

*Lockwood, 1999*). Briefly, biotic homogenization occurs when species are introduced to new areas, reducing the distinctness between source and site of introduction. Unsurprisingly, environment is a significant driver of tree community similarity between cities, but this association is stronger for naturally occurring (rather than introduced) trees (*Figure 4D*).

These data have been collected over many years by urban foresters, citizen scientists, consulting firms, and other interested parties; here, we could not evaluate each city's accuracy at species identification and location determination. Likewise, we could not fully control for different sampling schemes and sampling efforts (but see 'Materials and methods'). Future work could deploy tree experts to randomly resample trees in each city and compare the identification to that in our dataset.

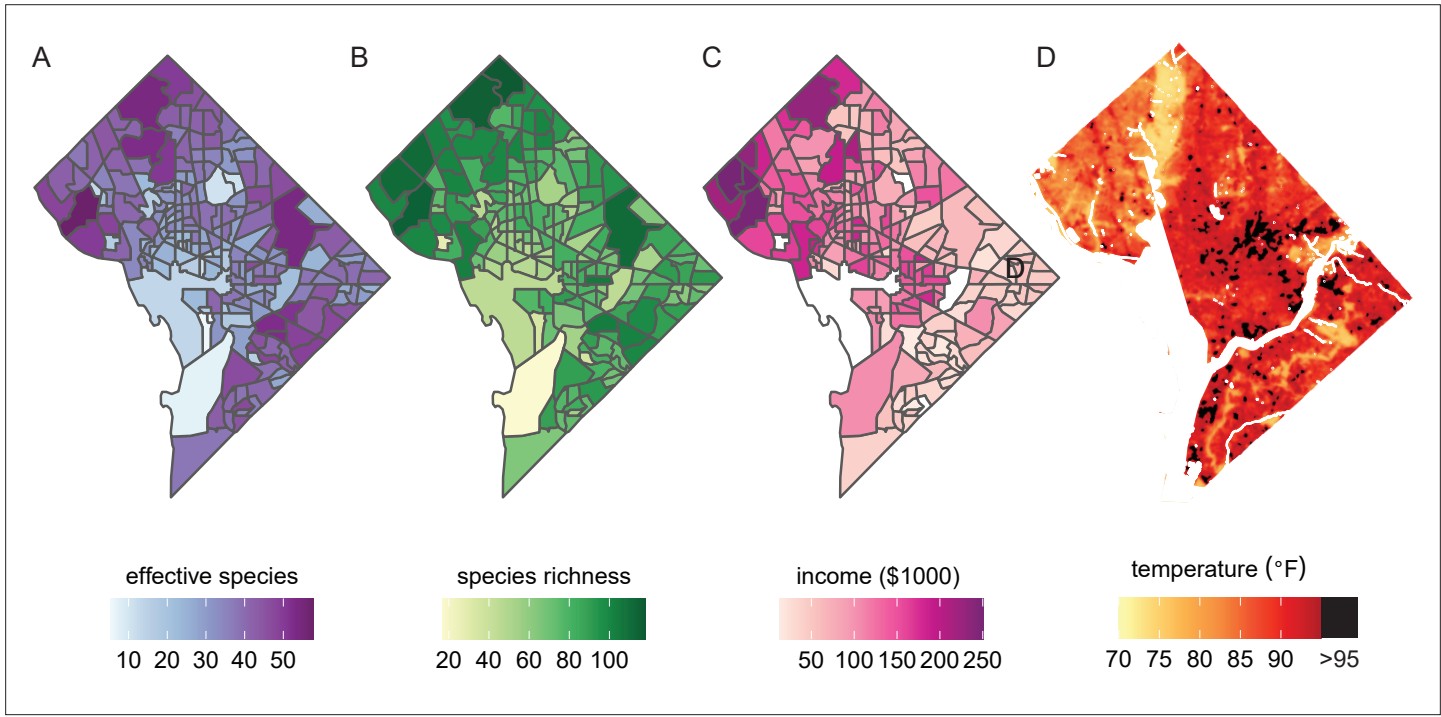

**Figure 5.** Future analyses could combine this city trees data with social, demographic, or physical variables (including income and urban heat islands). Here, we plot different variables for Washington, DC, showing qualitative concordance between (**A, B**) measures of species diversity, (**C**) household income, and (**D**) the location of urban heat islands. (**A**) Effective species count is highest in the northwest and varies by census tract from 7 species to 54 species (median = 35 species). (**B**) Species richness is also highest in the northwest and varies by census tract from 17 species to 118 species (median 77 species). (**C**) Median household income is highest in the northwest, the region which overlaps substantially with the most biodiverse city tree communities. (**D**) Land surface temperatures in July 2018 are plotted to show the spatial location of the highest temperatures, including urban heat islands with temperatures >95°F. Source data are *Figure 1—source data 1* and open-access data available from the US Census and the DC Open Data Portal (see 'Materials and methods') and source code is *Source code 2*.

## Future analyses: socioeconomics, demographics, the physical environment, and citizen-science species identification

Beyond the analyses demonstrated above, our dataset could also be combined with social, economic, and physical variables for new analyses (*Figure 5*). Simple maps of species diversity in the Washington, DC area (*Figure 5A, B*) show that high diversity qualitatively overlaps with high median household income (*Figure 5C*). In other words, not only do 'trees grow on money' (*Schwarz et al., 2015*), but they may be more diverse in richer areas (*Pedlowski et al., 2002*). Biodiverse green spaces improve mental health more than species-poor spaces (*Wood et al., 2018*) and likely have other synergistic benefits such as promoting more species diversity among birds and insects. Therefore, further analyses of city tree diversity by income, and other demographic factors, would be useful.

City trees cool urban temperatures (*Kong et al., 2014*) and clean the air, benefits which are not equitably distributed. For example, *Figure 5D* shows the location of heat islands in Washington, DC; urban heat islands can be cooled by planting city trees and increasing canopy cover (*Gartland, 2012*). The dataset herein could be combined with many physical variables for new analyses of how tree diversity and species compositions relate to temperature, air quality, and more.

Researchers could also analyze this city trees dataset in combination with other species diversity datasets gathered by citizen scientists. Members of the public frequently use popular phone applications to identify and document the location of birds, plants, insects, and more (*Bonnet et al., 2020*; *Chandler et al., 2017*). Future work could analyze whether a diverse city tree community correlates with a more biodiverse community of insects, birds, and even non-tree plants. Likewise, an analysis could consider whether the abundance of naturally occurring trees correlates with other important measures of ecosystem health (such as insect abundance). Since citizen-science datasets typically include exact location, future work could assess these trends over fine scales (e.g., within particular parks or in bounded neighborhoods) as well as across cities.

It would be useful to perform more refined analyses of clustering. For example, what is the biological significance of variation in cluster size (as determined by the hdbscan clustering algorithms)? The size and arrangement of the clusters themselves may be useful metrics. How clustered should we expect trees to be in both wild and urban settings? That is, what are our are null expectations? Further, researchers could apply network theory to predict how pest species would proliferate through each of these cities depending on the spatial arrangement of pest-sensitive trees.

Our study follows other impressive efforts to integrate and make inference from large sets of street tree inventories (e.g., *Kendal et al., 2014*; *Love et al., 2022*; *Ossola et al., 2020*). We concentrated our data collection on inventories with fine-scale tree locations and within a geographic context where plant species have been thoroughly characterized as introduced or naturally occurring, which allowed us to introduce two new approaches to this endeavor. First, we could evaluate how street tree diversity is spatially clustered within cities. Second, we could assess the influence of introduced versus naturally occurring tree species on driving tree community similarity between cities. Further, we also standardized data on tree health and developed new tools for analyzing datasets of urban forests. We anticipate that many further analyses of street tree inventories are yet to come.

## Conclusion

Humans consciously control urban ecosystems, in part by selecting and planting city trees. We have an opportunity to design diverse, spatially heterogeneous city tree communities with fewer introduced species—thereby building resilience against climate change (*Roloff et al., 2009*), avoiding pest/pathogen outbreaks (*Laćan and McBride, 2008*), improving human's mental and physical health (*Fuller et al., 2007*), and providing richer habitat for non-human animals (*Burghardt et al., 2009*; *Burghardt et al., 2010*; *Gallo and Fidino, 2018*; *Parsons et al., 2018*). We should use green decision-making to forge a path toward a sustainable urban future.

## Materials and methods

### Data acquisition

We limited our search to the 150 largest cities in the USA by census population.

To acquire raw data on street tree communities, we used a search protocol on both Google and Google Datasets Search (https://datasetsearch.research.google.com/). We first searched the **city**

**name** plus each of the following: **street trees**, **city trees**, **tree inventory**, **urban forest**, and **urban canopy** (all combinations totaled 20 searches per city, 10 each in Google and Google Datasets Search). We then read the first page of google results and the top 20 results from Google Datasets Search. If our search produced a city by the same name but in the wrong state, we redid the 20 searches adding the state name. If no data were found, we contacted a relevant state official via email or phone with an inquiry about their street tree inventory. Datasheets were received and transformed to CSV format (if they were not already in that format). We received data on street trees from 64 cities. One city, El Paso, had data only in summary format and was therefore excluded from analyses .

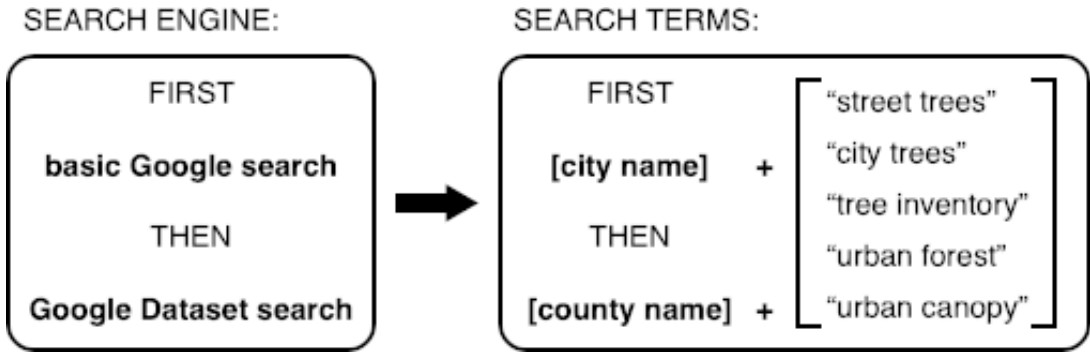

**Scheme 1.** Dataset search pipeline.

## Data cleaning

All code is in the zipped folder *Source code 1*. Before cleaning the data, we ensured that all reported trees for each city were located within the greater metropolitan area of the city (for certain inventories, many suburbs were reported—some within the greater metropolitan area, others not).

First, we renamed all columns in the received CSV sheets, referring to the metadata and according to our standardized definitions (*Supplementary file 1*). To harmonize tree health and condition data across different cities, we inspected metadata from the tree inventories and converted all numeric scores to a descriptive scale including 'excellent', 'good', 'fair', 'poor', 'dead', and 'dead/dying'. Some cities included only three points on this scale (e.g., 'good', 'poor', 'dead/dying') while others included five (e.g., 'excellent', 'good', 'fair', 'poor', 'dead').

Second, we used *pandas* in Python (*McKinney, 2011*) to correct typos, non-ASCII characters, variable spellings, date format, units used (we converted all units to metric), address issues, and common name format. In some cases, units were not specified for tree diameter at breast height (DBH) and tree height; we determined the units based on typical sizes for trees of a particular species. Wherever diameter was reported, we assumed it was DBH. We standardized health and condition data across cities, preserving the highest granularity available for each city. For our analysis, we converted this variable to a binary (see 'Condition and health'). We created a column called 'location_type' to label whether a given tree was growing in the built environment or in green space. All of the changes we made, and decision points, are preserved in *Source code 1*.

Third, we checked the scientific names reported using *gnr_resolve* in the R library *taxize* (*Chamberlain and Szöcs, 2013*), with the option *Best_match_only* set to TRUE (*Source code 1*). Through an iterative process, we manually checked the results and corrected typos in the scientific names until all names were either a perfect match ($N$ = 1771 species) or partial match with threshold greater than 0.75 ($N$ = 453 species). BGS manually reviewed all partial matches to ensure that they were the correct species name, and then we programmatically corrected these partial matches (e.g., *Magnolia grandifolia*—which is not a species name of a known tree—was corrected to *Magnolia grandiflora*, and *Pheonix canariensus* was corrected to its proper spelling of *Phoenix canariensis*). Because many of these tree inventories were crowd-sourced or generated in part through citizen science, such typos and misspellings are to be expected.

Some tree inventories reported species by common names only. Therefore, our fourth step in data cleaning was to convert common names to scientific names. We generated a lookup table by summarizing all pairings of common and scientific names in the inventories for which both were reported. We

manually reviewed the common to scientific name pairings, confirming that all were correct. Then we programmatically assigned scientific names to all common names (*Source code 1*).

Fifth, we assigned "native status" to each tree through reference to the Biota of North America Project (*Kartesz, 2018*), which has collected data on all native and non-native species occurrences throughout the US states. Specifically, we determined whether each tree species in a given city was naturally occurring in that state, introduced to that state, or that we did not have enough information to determine nativity (for cases where only the genus was known).

Sixth, some cities reported only the street address but not latitude and longitude. For these cities, we used the OpenCageGeocoder (https://opencagedata.com/) to convert addresses to latitude and longitude coordinates (*Source code 1*). OpenCageGeocoder leverages open data and is used by many academic institutions (see https://opencagedata.com/solutions/academia).

Seventh, we trimmed each city dataset to include only the standardized columns we identified in *Supplementary file 1*.

After each stage of data cleaning, we performed manual spot checking to identify any issues.

## Environmental variables

We retrieved WorldClim data on 19 bioclimatic variables using the *getData* function in package *raster* (*Hijmans and Etten, 2012*) with parameters var="bio" and res = 2.5. We used resolution = 2.5°, and as a sensitivity test we confirmed that these environmental values were significantly correlated with the same values at 0.5° resolution. We gathered climate variables for each city by extracting the grid cell closest to the latitude and longitude of each city in our dataset, and then we performed a PCA on the environmental variables.

## Species diversity

We calculated effective species counts (the exponent of the Shannon–Weiner index) as our measure of species diversity because it incorporates both richness (number of species) and evenness (distribution of those species; *Kendal et al., 2014*), and because it is a metric that behaves naturally and intuitively in comparisons between species communities (*Jost, 2006*). Effective species count is calculated as shown in *Equation 1*, where $n$ is the number of species present and $p_i$ is the frequency of a species $i$.

$$e^{\sum_{i=1}^{n} - p_i \ln(p_i)} \tag{1}$$

To determine what environmental and sociocultural factors drive species diversity (dependent variable: effective species count), we used the *olsrr* package in R (*Hebbali and Hebbali, 2017*) to compare AIC and adjusted $R^2$ values for all possible models incorporating the following independent variables: environmental PCA1, environmental PCA2, environmental PCA1 × environmental PCA2, city age, tree city USA (whether or not a city was designated as a tree city USA), city age × tree city USA, and the log-transformed number of trees in a given city.

Throughout our analyses, it was necessary to control for different sample sizes (and different, but unknown, sampling efforts across cities). To do so, we relied on the rarefaction/extrapolation methods developed by Chao and colleagues (*Chao et al., 2015*; *Chao et al., 2014*; *Chao and Jost, 2012*) and implemented through the R software package *iNext* (*Hsieh et al., 2016*). In short, these methods use statistical rarefaction and/or extrapolation to generate comparable estimates of diversity across populations with different sampling efforts or population sizes, alongside confidence intervals for these diversity estimates. *iNext* performs these tasks for Hill numbers of orders $q = 0$, 1, and 2. We used two techniques in *iNext* to allow for comparisons across cities (and between parks and urban areas within cities). First, we generated asymptotic diversity estimates for each; second, we generated diversity estimates for a given standardized population size. For our diversity analyses, the standardized population size we used was 37,000 trees (the rounded median of all cities). For analyses of the diversity of naturally occurring trees, we used a standardized population size of 10,000 trees (the rounded median across cities). For comparisons of the diversity between park and urban areas in a city, we used the smaller of the two population sizes (park or urban). In all cases, we also recorded confidence estimates and plotted rarefaction/extrapolation curves (*Figure 2—source data 1*).

To control for variation in how uniformly trees were sampled across a city's geographic range, we developed a procedure to score each city's spatial coverage (see 'Spatial structure').

We identified the best-fitting model, and then repeated our analysis under six sensitivity conditions to control for differences in population size, sampling effort, spatial coverage, and sample coverage. Our sensitivity analyses were as follows: first, with independent variable = effective species as calculated for a given population of 37,000 trees ; second, independent variable = the asymptotic estimate of the effective species number for that city as calculated using *iNext*; third, the raw effective species number; fourth, excluding cities with fewer than 10,000 trees; fifth, excluding cities with <50% spatial coverage; sixth, excluding cities with <0.995 sample coverage as calculated by *iNext*. For the fourth, fifth, and sixth models, the independent variable was effective species for a standardized population size of 37,000 trees.

We report statistics in *Supplementary file 2*.

## Spatial structure

We wanted to quantify the degree to which trees were spatially clustered by species within a city rather than randomly arranged. To do so, we first clustered all trees within each city using hierarchical density-based spatial clustering through the *hdbscan* library in Python (*McInnes et al., 2017*). HDBSCAN, unlike typical methods such as '*k* nearest neighbors', takes into account the underlying spatial structure of the dataset and allows the user to modify parameters in order to find biologically meaningful clusters. For city trees, which are often organized along grids or the underlying street layout of a city, this method can more meaningfully cluster trees than merely calculating the meters between trees and identifying nearest neighbors (which may be close as the crow flies but separated from each other by tall buildings). In particular, using the Manhattan metric rather than Euclidean metrics improves clustering analysis in cities (which tend to be organized along city blocks). For further discussion of why *hbdscan* is preferable to other clustering metrics, see *Berba, 2020*; *Leland et al., 2016*; *McInnes et al., 2017*.

We converted latitude and longitude values within a city to their planar projection equivalents (in Universal Transverse Mercator [UTM]) using the *from_latlon* function in Python package *UTM* (*Bieniek et al., 2016*). In total, we had *N* = 59 cities with spatial information about their trees.

We then clustered all the trees in a given city using *HDBSCAN* with parameters min_cluster_size = 30, min_samples = 10, metric = 'manhattan', cluster_selection_epsilon = 0.0004, cluster_selection_method = 'eom'; we arrived at these parameters through trial and error with a sample set of cities.

Once we had all trees in a city assigned to spatial clusters (or, for trees far from the clusters, notated as 'noise' and eliminated from further analysis), we used a bootstrapping method to quantify the degree of homogenization within spatial clusters. For each cluster of trees (e.g., a cluster of 120 trees in Pittsburgh, PA) we (1) calculated the observed effective species number; (2) we randomly resampled 120 trees from Pittsburgh's entire 45,703-tree-dataset and calculated the effective species number of that random group of 120 trees; (3) we repeated step (2) 500 times; (4) we recorded the mean, median, and interquartile range of effective species counts from those 500 samples; and (5) we divided the expected effective species (median effective species count from all 500 samples) by the observed effective species count in the actual spatial cluster of 120 trees. The resulting value therefore quantifies the degree to which a spatial cluster is a random set of that city's tree species (values close to 100%) or a nonrandom set of same-species clusters (values less than 100%).

Cities varied in how uniformly trees were sampled across a city's geography. To control for this variation, we generated a 'spatial coverage' score using the following procedure. First, we divided each city into grid cells of 0.005° latitude by 0.005° longitude, excluding water features and truncating grid cells by the city's borders, using the R packages rgdal (*Bivand et al., 2015*) and raster (*Hijmans and Etten, 2012*; *Hijmans et al., 2013*). Second, we counted the number of trees in each grid cell. Third, because some grid cells were smaller in terms of actual area (e.g., because some grid cells were located at the edge of a city, and because degrees do not translate consistently to m$^2$), we calculated the adjusted number of trees per grid cell (raw number of trees × grid cell area/(maximum grid cell area)). Fourth, we calculated the percent of grid cells with no trees as well as the skew and kurtosis of adjusted number of trees in all occupied cells (using functions from R package *moments*, *Komsta and Novomestky, 2015*). Fifth, we plotted all cities with trees assigned to grid cells and saved the raw and summary spatial coverage data (*Figure 4—source data 3*).

## Nativity Status

To determine whether a tree was introduced or naturally occurring ("native") in the state in which it appeared, we referred to the state-specific lists of native species from the Biota of North America Project. Each tree species was therefore coded as naturally_occurring, introduced, or no_info. Some tree records included only genus-level data, which was coded as 'no_info'.

We performed beta regression models with a logit link function using the package *betareg* in R (*Zeileis et al., 2019*), with percent naturally occurring trees in a given city as the dependent variable. We assumed the precision parameter $\phi$ did not depend on any regressors. We started with a model incorporating only environmental variables, based on the substantial evidence that climate impacts the diversity of naturally occurring species, and then added one variable at a time to determine whether the additional variables improved the model's performance (tested through the *lrtest*() function from the package *lmtest*, *Hothorn et al., 2015*). The best model incorporated the following dependent variables: environmental PCA1, environmental PCA2, log(number trees), and city age with no interaction terms.

We reran the models under four sensitivity tests to ensure that sampling effort, spatial coverage, sample size, and outliers did not impact our results. First, we identified and removed the outliers Honolulu, HI and Miami, FL. Second, we excluded all cities with fewer than 10,000 trees. Third, we excluded all cities with <50% spatial coverage. Fourth, we excluded all cities with <0.995 sample coverage as estimated in the *iNext* software package.

## Condition and health

We asked whether a tree's condition within a given city was correlated with size (DBH), location type (whether in the built environment or in green space such as a park), and nativity status. Fifteen cities had two or more of these variables with adequate sample sizes, and we ran separate logistic regression models by city because cities do not always score condition on comparable scales. We coded tree condition as a binary variable, where 'excellent', 'good', or 'fair' condition trees were coded as 1 and 'poor', 'dead', and 'dead/dying' trees were coded as 0. We used function *glm2*() in the R package *glm2* (*Marschner, 2011*), and for each model determined whether it was a better fit than an empty model. We calculated odds ratios, confidence intervals, and p values (see *Supplementary file 4*).

## Similarity between tree communities

How similar are species compositions across cities? For $N$ = 1953 city–city comparisons of street tree communities, we could calculate weighted measures of similarity because we had frequency data. We calculated similarity scores for the entire tree population, the naturally occurring trees only, and the introduced trees only. We used chi-square distance metrics on species frequency data, and we controlled for different population sizes (and potentially, sampling efforts) between cities by subsampling the larger city 50 times to match the smaller city's tree population size and calculating average metrics. In this manner, we controlled for differences in sample size. Chi-square similarity was calculated as in *Equation 2*, where $n$ is the total number of species present in either city, $x$ and $y$ are vectors of species frequencies for the two cities being compared, and for each species $i$, $x_i$ is the frequency of that species in city $x$ and $y_i$ the frequency of the same species in city $y$. Chi-square similarity is one minus the chi-square distance.

$$1 - \frac{1}{2} \sum_{i=1}^{n} \frac{(x_i - y_i)^2}{(x_i + y_i)} \tag{2}$$

We calculated environmental similarity as one minus the normalized euclidean distance in our PCA plot of environmental variables.

To determine whether city species similarity was driven by naturally occurring species, introduced species, or neither, we performed a two-sample paired t.test using the function *t.test* in R between the naturally occurring species chi-squared similarity scores and the all-species chi-squared similarity scores. Because the variables were not perfectly normally distributed (although they were even and symmetric), we also performed a non-parametric Wilcoxon signed-rank test. We plotted a histogram of the difference between each pair of city's chi squared scores for (1) all species and (2) naturally occurring species only.

To determine whether the environment was a stronger driver of naturally occurring species communities versus all species communities, we compared correlation scores. Specifically, we used the function *cor.test* in R to calculate the Pearson's product-moment correlation between chi-squared similarity and environmental similarity for (1) naturally occurring species only and (2) all species. We compared the all-species-environment correlation to the naturally occurring-species-environment by calculating Pearson and Filon's z using the *cocor* package in R (*Diedenhofen and Musch, 2015*) for two overlapping correlations based on dependent groups (calculation takes into account correlation between chisq_native and chisq_all, among other things).

### Income and urban heat islands

To demonstrate the value of our dataset for analyses of social, economic, and physical variables, we mapped several such variables for Washington, DC using packages *raster* (*Hijmans and Etten, 2012*), *sf* (*Pebesma, 2018*), and *tidycensus* (*Walker et al., 2021*) in R. First, we split our trees data by census tract and mapped species richness and effective species count within each tract; next, we extracted median household income data and plotted it for each census tract (*Walker, 2022*). Finally, we downloaded LANDSAT data on surface temperatures in DC for July 2018 from the DC Open Data portal (https://opendata.dc.gov/documents/land-surface-temperature-july-2018/explore; CC-BY-4.0) and plotted this, marking heat islands (temperature >95°F) in black (*Jolly, 2019*).

### Acknowledgements

We are so grateful to the innumerable citizen-foresters, trained arborists, and city government officials who have worked tirelessly to make this data available. For their help with this specific project, we wish to thank Andrew Pineda (Huntington Beach, CA), Bailey Patterson (Overland Park, KS), Brian Liberti (Rochester City, NY), Dan Buckler (WI), Daniel Hickey (Durham, NC), David Wrights (Oklahoma City, OK), Donna Davis (CO), Erik Dihle (Baltimore, MD), Gary Farris (Wichita, KS), Glenn Slaton (Durham, NC), Gretchen Erickson (Huntington Beach, CA), Jane Gregory and Maria Repass (Orange County, FL), John Saylor (Lexington, KY), Joran Viers (Albuquerque, NM), Kasey Krause (Knoxville, TN), Kevin Wilde and Bill Williams (Amarillo, TX), Matt Stull (Santa Rosa, CA), Nathan Randolph (Baltimore, MD), Rachot Moragraan (Garden Grove City, CA), Randy Menzel (Huntington Beach, CA), Russell Calhoun Jr. (Houston, TX), Shane D. McQuillan (Des Moines, IA), Terri Bladow (Santa Rosa, CA), The City of Saint Louis Forestry Division (St. Louis, MO), Steven Ashley (Louisville, KY), Todd Hayes (Greensboro, NC), and many other city government officials not named here. For the data for Hawaii, we wish to thank Heather McMillen, Wai Lee, and Terri-Ann Koike, and we wish to state that data used to help generate this report has been collected by the Citizen Forester Program; a collaborative project of the State and Private Forestry branch of the USDA Forest Service, Department of Agriculture, Region 5; the Kaulunani Urban and Community Forestry Program of the DLNR Division of Forestry and Wildlife; and Smart Trees Pacific. We would also like to thank Dan Utter, Nina Yancy, members of Sönke Johnsen's lab ,and attendees at Botany 2021 for useful feedback. This product uses the Census Bureau Data API but is not endorsed or certified by the Census Bureau. Funding: Stanford Science Fellowship (DEM) NSF Postdoctoral Research Fellowships in Biology PRFB Program, grant 2109465 (DEM) Theodore H Ashford Graduate Fellowship in the Sciences (DEM) Department of Defense, Army Research Office, National Defense Science and Engineering Graduate NDSEG Fellowship, 32 CFR 168 a (DEM) National Science Foundation NSF Evolution, Ecology, Environment (E3) Research Experience for Undergraduates REU program, award number 1757780 (BFA) The Franklin Delano Roosevelt Foundation Summer Research Grant (HK).

## Additional information

### Funding

| Funder | Grant reference number | Author |
| --- | --- | --- |
| National Science Foundation | Postdoctoral Research Fellowships in Biology 2109465 | Dakota E McCoy |

| Funder | Grant reference number | Author |
|---|---|---|
| National Science Foundation | Research Experience for Undergraduates (REU) 1757780 | Bulent Furkan Atahan |
| Stanford University | Science Fellowship | Dakota E McCoy |
| The Franklin Delano Roosevelt Foundation | Summer Research Grant | Hana Kiros |

The funders had no role in study design, data collection, and interpretation, or the decision to submit the work for publication.

### Author contributions
Dakota E McCoy, Conceptualization, Data curation, Software, Formal analysis, Supervision, Funding acquisition, Investigation, Visualization, Methodology, Writing – original draft, Project administration, Writing – review and editing; Benjamin Goulet-Scott, Conceptualization, Data curation, Software, Formal analysis, Supervision, Funding acquisition, Validation, Investigation, Methodology, Writing – original draft, Project administration, Writing – review and editing; Weilin Meng, Data curation, Software, Formal analysis, Funding acquisition, Investigation, Methodology, Writing – review and editing; Bulent Furkan Atahan, Hana Kiros, Data curation, Funding acquisition, Investigation, Writing – review and editing; Misako Nishino, John Kartesz, Resources, Data curation, Funding acquisition, Investigation, Writing – review and editing

### Author ORCIDs
Dakota E McCoy (iD) http://orcid.org/0000-0001-8383-8084
Benjamin Goulet-Scott (iD) http://orcid.org/0000-0003-2004-6586

### Decision letter and Author response
Decision letter https://doi.org/10.7554/eLife.77891.sa1
Author response https://doi.org/10.7554/eLife.77891.sa2

---

## Additional files

### Supplementary files
• Supplementary file 1. Here, we define the standardized columns used herein.

• Supplementary file 2. Here, we report the results of linear regression models analyzing diversity of the tree communities. We ran six sensitivity analyses as follows. In Model 1, the independent variable was effective species as calculated for a given population of 37,000 trees ('effective species for a standardized population size'). In Model 2, the independent variable was the asymptotic estimate of the effective species number for that city as calculated using *iNext*. In Model 3, the independent variable was the raw effective species number. In Model 4, we excluded cities with fewer than 10,000 trees. In Model 5, we excluded cities with <50% spatial coverage. In Model 6, we excluded cities with <0.995 sample coverage as calculated by *iNext*. For Models 4—6, the independent variable was effective species for a standardized population size of 37,000 trees. We use $p = 0.05$ as our threshold for significance; it is important to note that we do not apply a Bonferroni multiple-testing correction, because these are sensitivity analyses repeating the original test (rather than new, but statistically related, hypothesis tests).

• Supplementary file 3. Here, we report the results of beta regression models with independent variable percent naturally occurring trees (percent of trees that occur naturally in that region rather than being introduced). We ran five models as sensitivity analyses to check for effects of population size, sample completeness, and spatial coverage, In Model 1, we include all cities. In Model 2, we excluded the outlier cities Honolulu, HI and Miami, FL. In Model 3, we excluded cities with fewer than 10,000 trees. In Model 4, we excluded cities with <50% spatial coverage. In Model 5, we excluded cities with <0.995 sample coverage as calculated by *iNext*. We use $p = 0.05$ as our threshold for significance; it is important to note that we do not apply a Bonferroni multiple-testing correction, because these are sensitivity analyses repeating the original test (rather than new, but statistically related, hypothesis tests).

• Supplementary file 4. We ran logistic regression models to identify correlations between condition and (1) tree size, (2) tree location, and (3) whether a tree was naturally occurring or introduced. For

size, smaller trees (lower diameter at breast height) tended to have better condition (12 of 18 cities). For location type, trees in the built environment tended to have better condition than those in parks (four of eight cities). For native status, results were mixed (naturally occurring trees had no difference in condition for 6 of 19 cities, worse condition in 7 cities, and better condition in 6 cities). Significant odds ratios and models are marked with *p < 0.05, **p < 0.005, ***p < 0.0005, or NS: p ≥ 0.05.

• Supplementary file 5. Loadings for the environmental principal component analysis in *Figure 4A*. The Bioclim variables having to do with precipitation are negatively correlated with PCA1 and positively correlated with PCA2. When PCA1 is high and PCA2 is low, precipitation is higher. Likewise, when PCA1 is low and PCA2 is high, precipitation is lower. For example, refer to loadings for Annual_Precip and Precip_Driest_Month.

• Transparent reporting form

• Source code 1. Code_for_Data_Cleaning.zip. This zipped file includes all code sheets in Python and R, and instructions, for the full data cleaning procedure (see 'Materials and methods', Data Cleaning).

• Source code 2. Code_for_Analysis_and_Plotting.zip. This zipped file includes all code used to analyze and plot the results reported in this paper.

• Source code 3. Calculate_Effective_Species_Excel_Tool.xlsx. We developed an Excel Spreadsheet which calculates effective species counts, a robust measure of species diversity, from a list of all trees (each row is an individual tree).

• Source code 4. Check_Native_Status_of_Species_Excel_Tool.xlsx. This Excel Workbook allows readers to input their list of species, select a state, and receive a corresponding list of whether or not each species is native to that state (based on BONAP designations).

## Data availability

All data and code are available in the main text or the supplementary materials. The datasheets of city tree information from 63 cities Figure 1- source data 1 (63 .csv files) have been uploaded to Dryad: https://doi.org/10.5061/dryad.2jm63xsrf.

The following dataset was generated:

| Author(s) | Year | Dataset title | Dataset URL | Database and Identifier |
|---|---|---|---|---|
| McCoy DE, Goulet-Scott B, Meng W, Atahan B, Kiros H, Nishino M, Kartesz J | 2022 | A dataset of 5 million city trees from 63 US cities: species, location, nativity status, health, and more | https://dx.doi.org/10.5061/dryad.2jm63xsrf | Dryad Digital Repository, 10.5061/dryad.2jm63xsrf |

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
