## [Editor Report]

This paper will be of interest to urban foresters, ecologists, and planners. It provides a large new dataset of city tree communities across US cities, which may ignite new studies on city biodiversity and ecosystem services. It contains clear descriptions about the data processing and structures, and the potential uses of the data.

---

## [Decision Letter]

**Decision letter after peer review:**

Thank you for submitting your article "A dataset of 5 million city trees: species clustering and climate effects in urban forests" for consideration by *eLife*. Your article has been reviewed by 3 peer reviewers, one of whom is a member of our Board of Reviewing Editors, and the evaluation has been overseen by a Reviewing Editor and Meredith Schuman as the Senior Editor. The reviewers have opted to remain anonymous.

Essential revisions:

In your revision, please respond point-by-point to these essential revisions agreed to in the consultative review process. The original reviews are also provided below for your reference.

1. Sample completeness and representativeness

The tree inventories varied significantly regarding the number of records (214-720,140 trees per city). The variation can be due to the actual variation of tree abundances in studied cities or incomplete or biased inventories. Please add a column in the meta data indicating the sampling effort or completeness and if there were any sampling bias in the case of incomplete samples. Please also add clarifications about sample completeness and representativeness in the main text.

2. If there are incomplete or biased samples, please assess how these would affect the validities of the analyses and conclusions, especially those related to α and β diversity.

a) Please note that effective species number (and other diversity indices) is dependent on sample coverage (i.e., proportions of species were sampled) and sample size (i.e., area or abundance), so that the comparisons of α diversity can be unsupported if the samples are incomplete, there are many rare species with few individuals in cities, or sampling efforts vary between common and rare species. In these cases, it is necessary to standardize biodiversity measures before comparisons on the basis of sample coverage or sample size, using rarefaction-extrapolation approaches developed by Anne Chao and colleagues (Chao and Jost. 2012. Ecology 93:2533-2547; Chao et al. 2014. Ecological Monographs 84:45-67.). We note that the analyses on the relationships between biodiversity and environment, which included log-transformed tree abundance as a covariate, did consider the dependence on sample size in a way. However, effective species number may not change with tree abundance in a log-linear way, and some other relevant analyses (e.g., comparisons between parks and urban settings) did not have proper standardization. It should also be noted that one city with a small sample size can have a high sample coverage, and sample standardization approaches suggested above originally developed for species-rich ecosystems (especially for those with many rare species) may not be completely suitable or necessary for city trees planted by humans, so if the authors have arguments for why the dependence of sample coverage or sample size do not apply for a particular question, please make that clarification.

b) If there are apparent sample biases or very low sample coverages, additional steps other than rarefaction-extrapolation analyses may be required, for example, performing sensitivity analyses by dropping unrepresentative samples.

3. More information about data processing is required to increase users' confidence in the quality of data. How did you derive locations from the address? How did you validate the location accuracy? How did you harmonize the health evaluation results from different investigators? How did you evaluate the reliability of the data set (e.g., species identification) gathered from different times by different investigators? It may be helpful to assign a quality label to each record based on specific standards.

4. Please clarify the relative strength and weakness of the new data relative to the Global Urban Tree Inventory developed by Ossola et al., (Global Ecology and Biogeography 2020: 29:1907-1914) and records in Global Biodiversity Information Facility.

5. Additional context/conceptual underpinning the clustering analysis would be great. How did the analyses find the biologically meaningful clusters and recognize that a tall building exists or not to separate trees? What can we infer from variation in the sizes of clusters?

6. Please note the differences in the definitions of several critical concepts used in the paper, including city tree vs. urban forest, biodiversity vs. species diversity vs. effective species number. They are relevant but not identical, so caution is needed when using them interchangeably.

7. Some supplemental tools did not work or contain instructions when downloaded in the way. The excel supplemental tools need directions on the form itself to make them usable.

8. Please clarify your research questions or objectives in the Introduction section.

9. Please avoid having a colon in the title.

*Reviewer #2 (Recommendations for the authors):*

Line 46-49. The first half of the sentence used city trees, while the second half used urban forests. These two terms are not equal unless the authors give their own definition of the urban forest.

Line 53-54. Same as above.

Line 58. Precise is not a suitable term since coordinates of trees in some cities were derived from addresses.

Line 64. What does "a metric will depend on biodiversity" mean? Do you mean species diversity? It is not equal to biodiversity.

Line 71. It is an overstatement that the authors take a first step toward enabling the design of rich, heterogeneous ecosystems built around urban forests. There are already some global and regional databases of urban tree species. Each has its own strengths.

Line 77. Here the authors stated that the database was compiled from street tree inventories. If this is true, they cannot use city trees or urban forests to refer to their collected data because street trees are only a small part of city trees or urban forests.

Line 97. The huge variance in the number of tree records in different inventories is problematic. It indicates that some inventories were incomplete, e.g., 214 trees in Phoenix, AZ. No meaningful comparison can be made without accounting for this problem. One cannot attribute the difference in findings to influencing factors or incomplete inventories. The authors can run a completeness analysis to sift out cities that did not have complete inventories before further analysis.

Line 117. Effective species count is for measuring diversity, not biodiversity. Please see the Lou Jost's website as listed in the paper (Line 369-370).

Line 210-211. What is the relationship between considering hardiness when selecting tree species and tree health metrics? The health metrics used by the authors are rather general, not designed specifically for measuring a tree specie's hardiness.

Line 256-259. The interpretation that heat islands are more common in species-poor areas of Washington, DC is a overstretch. There are so many confounding factors, including anthropogenetic heat sources, urban layout, and the tree cover as mentioned by the authors.

Line 271. The statement should be rephrased. The human can never exercise precise control of urban forests. Many biotic and abiotic factors that we cannot control impact urban forests. The climate factors discussed by the authors are a good example.

Line 313-356. Tree inventories can be generated using different sampling schemes, e.g., census or probability-based sampling methods. I assumed that the authors had the details in the supplementary files. However, it will help readers if the authors can briefly introduce how these different types of inventories were processed in the main text, especially on these incomplete tree inventories.

Line 360. Please explain why the resolution of 2.5 degrees was used. 2.5 degrees roughly equal 250 km. Probably no single US city has such a dimension. WorldClim data are available at resolutions as high as 30 arcseconds (~1 km). It may be more suitable for studies on a city scale.

Line 482. Please explain why use temperature >95o F as the threshold value? The authors used land surface temperature derived from satellite data. It described the thermal patterns of urban surface, not urban heat island.

*Reviewer #3 (Recommendations for the authors):*

Not totally necessary, but may want to consider the using the terms "introduced" vs. "naturally occurring" species rather than “native” vs. “non-native”. There are some arguments from humanities folks about the history of the term native in colonialist narratives. Leaving it as is may distract from the very real biological argument that you are trying to make about the evolutionary history of locally interacting species leading to more interactions being supported by naturally occurring species than those that humans introduced more recently.

"…validate these tools in comparison to past methods…"- It is unclear to me from the text how validation was done.

Figure 4- what does the green line in the right panel of D represent- it isn't defined in the caption…

As also mentioned in the paper- you may want to consider specifying in your captions text etc that it is Shannon's effective species number you are using. Effective species can cover all hill numbers with q as a scaler (i.e. Simpson's effective species number is also an "effective species count")

The excel supplemental tools need directions on the form itself to make them usable

Data S2-

– Some "most_common_species" are a genus instead (i.e Santa Rosa, Detroit). What is going on.

– 4 columns are all N/As- why are they included?

Data S6 tool doesn't work for me. I pick my state and put in species and it doesn't work

Data S3 tool doesn't work either- just get a Div/0 error if you modify the species list.

Figure S2. "Effective species is a more nuanced metric of biodiversity than classic abundance-based measures". Define what classic abundance-based means. I think you mean the % of most common species which is I believe related to the Berger-Parker dominance Index. The confusion lies because for a community ecologist effective species number (i.e. exponent of Shannon's diversity) is a classic index that takes into account relative abundance as well as richness.

Posting R code to github as well would substantially improve the usability of the code for more sophisticated users and allow potential modifications as datasets are added.

Figure 5- "Land surface temperature in July 2018 shows that the highest temperatures, and 290 urban heat islands with temperatures > 95ºF, tend to overlap with less-richly-forested areas."- This figure does not show forest cover and I don't believe the DC data set includes every tree in urban forests so this statement does not seem supported by the analysis as currently completed

[Editors' note: further revisions were suggested prior to acceptance, as described below.]

Thank you for resubmitting your work entitled "Species clustering, climate effects, and introduced species in 5 million city trees across 63 US cities" for further consideration by *eLife*. Your revised article has been evaluated by Meredith Schuman (Senior Editor) and a Reviewing Editor.

The manuscript has been substantially improved. We appreciate all the efforts made by the authors. There are some remaining issues that need to be addressed, as outlined below:

1. L177: the numbers of effective species counts were inconsistent with those presented in the legend of Figure 2. Please make the correction.

2. Please provide the degrees of freedom (df) for the results of paired t-test in the legends of Figure 2, Figure 2—figure supplement 1, Figure 4.

3. L227-228: please clarify environmental PC1 and cite the PCA results (supplementary file 5) in the main text before introducing this result.

4. Figure 3: there is one city having a very large 95%CI, overlapping both the lines of zero and 100%. This case is very strange. How could this happen? How is possible that an observed effective species per cluster = 0?

5. L423: should be "what are our null expectations".

6. L1065: please clarify whether the effective species numbers were raw statistics or based on standardized population size.

7. L1107: should be in "Figure 4A".

---

## [Author Response]

Essential revisions:In your revision, please respond point-by-point to these essential revisions agreed to in the consultative review process. The original reviews are also provided below for your reference.1. Sample completeness and representativenessThe tree inventories varied significantly regarding the number of records (214-720,140 trees per city). The variation can be due to the actual variation of tree abundances in studied cities or incomplete or biased inventories. Please add a column in the meta data indicating the sampling effort or completeness and if there were any sampling bias in the case of incomplete samples. Please also add clarifications about sample completeness and representativeness in the main text.

This is a great point. We made two major changes (new analyses) and one minor change (new text clarifications) in response to this point. We further explain our changes in our response to your point #2 below.

First, we redid all of our diversity analyses using Chao et al’s methods of rarefaction and extrapolation in the R package *iNext*. This allowed us to add a column estimating sample completeness to our datasheet. We also redid the community comparison analysis (See response to #2 for further details).

Second, we designed a method to assess spatial coverage of each city as an indirect measure of sampling effort (as well as a direct measure of spatial completeness). By dividing each city into grid cells, counting the trees per unit area in each grid cell, we could then assign metrics of completeness (what percentage of grid cells were empty?) and bias (the skew and kurtosis of the numbers of trees in the occupied grid cells). Therefore, we re-ran our analyses with sensitivity measures whereby we excluded cities falling below 50% spatial completeness.

Compare Austin to Baltimore in Author response image 1 as an example.

**Author response image 1. sa2fig1:** 

Third, as a minor change, we added the following clarifying text to our manuscript:“For all analyses, when comparing diversity measures across different size scales, we applied rarefaction and extrapolation techniques using the R package *iNext* (See ‘Materials and methods’; Chao et al., 2015, 2014; Chao and Jost, 2012; Hsieh, Ma, and Chao, 2016) and performed sensitivity analyses excluding low-coverage cities.”

Finally, we added substantial new text to our Methods, new analyses, and new Supplementary files.

2. If there are incomplete or biased samples, please assess how these would affect the validities of the analyses and conclusions, especially those related to α and β diversity.a) Please note that effective species number (and other diversity indices) is dependent on sample coverage (i.e., proportions of species were sampled) and sample size (i.e., area or abundance), so that the comparisons of α diversity can be unsupported if the samples are incomplete, there are many rare species with few individuals in cities, or sampling efforts vary between common and rare species. In these cases, it is necessary to standardize biodiversity measures before comparisons on the basis of sample coverage or sample size, using rarefaction-extrapolation approaches developed by Anne Chao and colleagues (Chao and Jost. 2012. Ecology 93:2533-2547; Chao et al. 2014. Ecological Monographs 84:45-67.). We note that the analyses on the relationships between biodiversity and environment, which included log-transformed tree abundance as a covariate, did consider the dependence on sample size in a way. However, effective species number may not change with tree abundance in a log-linear way, and some other relevant analyses (e.g., comparisons between parks and urban settings) did not have proper standardization. It should also be noted that one city with a small sample size can have a high sample coverage, and sample standardization approaches suggested above originally developed for species-rich ecosystems (especially for those with many rare species) may not be completely suitable or necessary for city trees planted by humans, so if the authors have arguments for why the dependence of sample coverage or sample size do not apply for a particular question, please make that clarification.b) If there are apparent sample biases or very low sample coverages, additional steps other than rarefaction-extrapolation analyses may be required, for example, performing sensitivity analyses by dropping unrepresentative samples.

We’re grateful for this detailed comment, because we implemented all of your suggestions and the paper is much stronger as a result.

First, we redid all of our diversity calculations applying Chao’s rarefaction and extrapolation techniques through the R package *iNext*. Therefore, our summary datasheet now has many new columns to include the following values for each city:

Effective species number:

·Raw effective species numberAsymptotic estimate of effective species number with confidence intervalEstimate of effective species number for a given population size (37,000 trees– the median population size rounded to the nearest 1,000) with confidence interval

Species richness:

Raw species richness (number of species)Asymptotic estimate of number of species with confidence intervalEstimate of number of species for a given population size (37,000 trees– the median population size rounded to the nearest 1,000) with confidence interval

The same for the native-only population of trees in each city (e.g., not just raw number of effective number of native species but also the *iNext* estimates and confidence intervals)

Whether or not each of the values above was calculated using extrapolation or interpolation

Sample coverage estimates

Second, we re-ran our models testing for significant correlations between species diversity in a city and other factors (including climate), where we used the extrapolated / interpolated effective species numbers from *iNext*. Specifically, we found the best fit model, which included the following predictors: environmental PCA1, environmental PCA1:environmental PCA2, and whether or not a city was designated as a Tree City USA. Then, we ran this model under six sensitivity conditions, varying the independent variable and/or which cities we included based on completeness of their sample. Climate was still a significant correlate of diversity.

First, with independent variable = effective species as calculated for a given population of 37,000 trees ("effective species for a standardized population size");Second, independent variable = the asymptotic estimate of the effective species number for that city as calculated using *iNext*;Third, the raw effective species number;Fourth, excluding cities with fewer than 10,000 trees;Fifth, excluding cities with <50% spatial coverage;Sixth, excluding cities with <0.995 sample coverage as calculated by *iNext*.

For the fourth, fifth, and sixth models, the independent variable was effective species for a standardized population size of 37,000 trees.

Third, we redid our comparisons of tree populations in parks versus those in urban areas. Parks were still more diverse than urban areas.

Specifically, we used *iNext* to calculate diversity metrics based on the smaller of the two population sizes (park vs urban) to enable fair comparison for each city.

We reported comparison results for (i) raw effective species number, (ii) asymptotic estimate, and (iii) estimate for a given population.

In doing so, we eliminated Milwaukee from the comparison (it had only 28 trees recorded as being in an urban setting).

Fourth, we redid our pairwise comparisons of tree community composition between cities in order to account for different population sizes and sampling efforts. To do so, we randomly subsampled the larger city to make its population equal to the smaller city, calculated comparison metrics, and repeated this process 50 times. We report the average comparison metrics.

Our new Methods text is copied here for your convenience:

“Throughout our analyses, it was necessary to control for different sample sizes (and different, but unknown, sampling efforts across cities). To do so, we relied on the rarefaction / extrapolation methods developed by Chao and colleagues (Chao et al., 2015, 2014; Chao and Jost, 2012) and implemented through the R software package *iNext* (Hsieh et al., 2016). In short, these methods use statistical rarefaction and/or extrapolation to generate comparable estimates of diversity across populations with different sampling efforts or population sizes, alongside confidence intervals for these diversity estimates. *iNext* performs these tasks for Hill numbers of orders q = 0, 1, and 2. We used two techniques in *iNext* to allow for comparisons across cities (and between parks and urban areas within cities). First, we generated asymptotic diversity estimates for each; second, we generated diversity estimates for a given standardized population size. For our diversity analyses, the standardized population size we used was 37,000 trees (the rounded median of all cities). For analyses of the diversity of native trees, we used a standardized population size of 10,000 trees. For comparisons of the diversity between park and urban areas in a city, we used the smaller of the two population sizes (park or urban). In all cases we also recorded confidence estimates, and plotted rarefaction/extrapolation curves (DataS_Rarefaction_Plots.zip).

To control for variation in how uniformly trees were sampled across a city’s geographic range, we developed a procedure to score each city’s spatial coverage (see section Spatial Structure below).

We identified the best-fitting model, and then repeated our analysis under six sensitivity conditions to control for differences in population size, sampling effort, spatial coverage, and sample coverage. Our sensitivity analyses were as follows: first, with independent variable = effective species as calculated for a given population of 37,000 trees ("effective species for a standardized population size"); second, independent variable = the asymptotic estimate of the effective species number for that city as calculated using *iNext*; third, the raw effective species number; fourth, excluding cities with fewer than 10,000 trees; fifth, excluding cities with <50% spatial coverage; sixth, excluding cities with <0.995 sample coverage as calculated by *iNext*. For the fourth, fifth, and sixth models, the independent variable was effective species for a standardized population size of 37,000 trees.”

3. More information about data processing is required to increase users' confidence in the quality of data. How did you derive locations from the address? How did you validate the location accuracy? How did you harmonize the health evaluation results from different investigators? How did you evaluate the reliability of the data set (e.g., species identification) gathered from different times by different investigators? It may be helpful to assign a quality label to each record based on specific standards.

We derived locations from the addresses using an API service called OpenCage Geocoding (https://opencagedata.com/) to convert addresses into lat long coordinates. Open cage leverages open data (https://opencagedata.com/credits), and it is widely used by academics (see description here: https://opencagedata.com/solutions/academia). We added the following clarifying text:

“OpenCageGeocoder leverages open data and is used by many academic institutions (see https://opencagedata.com/solutions/academia).”

As you note, health is evaluated differently in different cities. We took three steps to address this.

First, we inspected metadata from city tree inventories to convert all assigned health scores to a descriptive scale including “excellent,” “good”, “fair”, “poor”, “dead”, and “dead/dying”. Some cities included only three points on this scale (i.e., “good”, “poor”, “dead/dying”) while others included five (e.g., “excellent,” “good”, “fair”, “poor”, “dead”).Second, for analysis, we converted this variable to a binary variable of where “excellent,” “good”, or “fair” condition trees were coded as 1 and “poor”, “dead”, and “dead/dying” trees were coded as 0. We consider these to be trustworthy standardized categories across cities.Third, because nonetheless cities likely had different internal criteria, we analyzed these results separately by city using the R function glm().

Therefore, we updated the Methods text describing this to read:

“To harmonize tree health and condition data across different cities, we inspected metadata from the tree inventories and converted all numeric scores to a descriptive scale including “excellent,” “good”, “fair”, “poor”, “dead”, and “dead/dying”. Some cities included only three points on this scale (e.g., “good”, “poor”, “dead/dying”) while others included five (e.g., “excellent,” “good”, “fair”, “poor”, “dead”). “

“We standardized health and condition data across cities, preserving the highest granularity available for each city. For our analysis, we converted this variable to a binary (see section Condition and Health).”

As for evaluating the reliability of the dataset, that is difficult. We gave much thought to this question and do not think we can come up with a way to quantify reliability easily. Therefore, we added the following disclaimer:

“These data has been collected over many years by urban foresters, citizen scientists, consulting firms, and other interested parties; here, we could not evaluate each city’s accuracy at species identification and location determination. Likewise, we could not fully control for different sampling schemes and sampling efforts (but see ‘Materials and methods’). Future work could deploy tree experts to randomly resample trees in each city and compare the identification to that in our dataset…”

4. Please clarify the relative strength and weakness of the new data relative to the Global Urban Tree Inventory developed by Ossola et al., (Global Ecology and Biogeography 2020: 29:1907-1914) and records in Global Biodiversity Information Facility.

Thank you for mentioning that excellent paper. Reviewer #2 noted that compared to the Ossola et al. (2020) paper, our dataset "“added information on spatial location, nativity statuses, and tree health conditions besides occurrences. The new information expands data usability and saves valuable time for researchers. The authors also make the tools available so others can use them to process their own data sets."

We added the following discussion:

“Our study follows other impressive efforts to integrate and make inference from large sets of street tree inventories (e.g. Kendal et al. 2014, Ossola et al. 2020). Concentrating our data collection on inventories with fine-scale tree locations and within a geographic context where plant species have been thoroughly characterized as introduced or naturally-occurring allowed us to introduce two new approaches to this endeavor. First, we could evaluate how street tree diversity is spatially clustered within cities. Second, we could assess the influence of introduced versus naturally-occurring tree species on driving tree community similarity between cities. Further, we also standardized data on tree health and developed new tools for analyzing datasets of urban forests. We anticipate that many further analyses of street tree inventories are yet to come.”

5. Additional context/conceptual underpinning the clustering analysis would be great. How did the analyses find the biologically meaningful clusters and recognize that a tall building exists or not to separate trees? What can we infer from variation in the sizes of clusters?

Compared to kmeans, and clustering methods that simply gather a point’s nearest neighbors regardless of the underlying spatial structure of the data, hdbscan takes into account the data’s underlying structure. It outperforms other clustering methods based on an intuitive understanding of what points should cluster together (see: https://hdbscan.readthedocs.io/en/latest/comparing_clustering_algorithms.html). For a basic explanation of HDBSCAN and why it works better than k-means, this blog post explains nicely: (https://towardsdatascience.com/a-gentle-introduction-to-hdbscan-and-density-based-clustering-5fd79329c1e8). Secondly, normally for clustering algorithms, we use the euclidean distance metric which measures distance like in a straight line. We used a manhattan distance algorithm which takes into account that cities are usually organized into blocks, so the distance between x and y are in right angles, rather than in a straight line (see Author response image 2). This leads to a more natural clustering that is suitable for cities.

Our text now reads:“We wanted to quantify the degree to which trees were spatially clustered by species within a city (rather than randomly arranged). To do so, we first clustered all trees within each city using hierarchical density based spatial clustering through the *hdbscan* library in Python (McInnes et al., 2017). *HDBSCAN*, unlike typical methods such as “k nearest neighbors”, takes into account the underlying spatial structure of the dataset and allows the user to modify parameters in order to find biologically meaningful clusters. For city trees, which are often organized along grids or the underlying street layout of a city, this method can more meaningfully cluster trees than merely calculating the meters between trees and identifying nearest neighbors (which may be close as the crow flies but separated from each other by tall buildings). In particular, using the Manhattan metric rather than Euclidean metrics improves clustering analysis in cities (which tend to be organized along city blocks). For further discussion of why hbdscan is preferable to other clustering metrics, see (Berba, 2020; Leland McInnes et al., 2016; McInnes et al., 2017).”

As for what we can conclude from the size of the clusters, this is complicated and would be an appropriate study of future work. Therefore, we added the following text:

“Researchers could also use this dataset to perform more refined analysis of clustering. For example, what is the biological significance of variation in cluster size (as determined by the hdbscan clustering algorithms)? The size and arrangement of the clusters themselves may be useful metrics. How clustered should we expect trees to be in both wild and urban settings? That is, what our are null expectations? Further, researchers could apply network theory to predict how pest species would proliferate through each of these cities (depending on the spatial arrangement of pest-sensitive trees).

6. Please note the differences in the definitions of several critical concepts used in the paper, including city tree vs. urban forest, biodiversity vs. species diversity vs. effective species number. They are relevant but not identical, so caution is needed when using them interchangeably.

Thank you for this note- we have edited our language throughout to make it more precise. For example, we replaced nearly all instances of “urban forest” with “city tree community” (except in the few cases where we were actually discussing the entire urban forest), and we replaced nearly all instances of “biodiversity” with diversity or “species diversity”.

7. Some supplemental tools did not work or contain instructions when downloaded in the way. The excel supplemental tools need directions on the form itself to make them usable.

We added instructions to both of our supplemental Excel Tools, and corrected errors in the Excel code so that the sheets should work properly now. Please let us know if they work for you now.

8. Please clarify your research questions or objectives in the Introduction section.

We added the following text to our introduction:

“In particular, we wanted to know whether local climatic conditions are associated with the species diversity of city tree communities, how species diversity was distributed in space within cities, and whether introduced tree species contribute to biotic homogenization among urban ecosystems.”

9. Please avoid having a colon in the title.

We have changed the title to:

“Species clustering, climate effects, and introduced species in more than 5 million city trees across 63 US cities”

Reviewer #2 (Recommendations for the authors):Line 46-49. The first half of the sentence used city trees, while the second half used urban forests. These two terms are not equal unless the authors give their own definition of the urban forest.

We have rewritten this sentence and provide definitions, as you suggest.

Line 53-54. Same as above.

We corrected this to “city plant life” and went through the document to replace all instances of “urban forest” with “city trees” in places where what we really meant was “city trees” (nearly all instances). In some cases we kept “urban forest” to refer to the broader topic of research and public interest, of which city trees are one important part.

Line 58. Precise is not a suitable term since coordinates of trees in some cities were derived from addresses.

We removed the word “precise”

Line 64. What does "a metric will depend on biodiversity" mean? Do you mean species diversity? It is not equal to biodiversity.

Yes, good catch- we changed this to “tree species diversity”

Line 71. It is an overstatement that the authors take a first step toward enabling the design of rich, heterogeneous ecosystems built around urban forests. There are already some global and regional databases of urban tree species. Each has its own strengths.

We agree. “First step” is not accurate (we certainly are not first!!). We changed it to our true aim, which is “help to enable”.

Line 77. Here the authors stated that the database was compiled from street tree inventories. If this is true, they cannot use city trees or urban forests to refer to their collected data because street trees are only a small part of city trees or urban forests.

Good catch- it was not street tree inventories but tree inventories (including many trees which are not alongside streets). We made this change.

Line 97. The huge variance in the number of tree records in different inventories is problematic. It indicates that some inventories were incomplete, e.g., 214 trees in Phoenix, AZ. No meaningful comparison can be made without accounting for this problem. One cannot attribute the difference in findings to influencing factors or incomplete inventories. The authors can run a completeness analysis to sift out cities that did not have complete inventories before further analysis.

We agree. We redid all our analyses applying rarefaction and extrapolation methods (referring to the body of work by Chao and colleague); in addition, we performed six sensitivity analyses where we excluded cities that did not meet different definitions of completeness

“…repeated our analysis under six sensitivity conditions to control for differences in population size, sampling effort, spatial coverage, and sample coverage. Our sensitivity analyses were as follows: first, with independent variable = effective species as calculated for a given population of 37,000 trees ("effective species for a standardized population size"); second, independent variable = the asymptotic estimate of the effective species number for that city as calculated using *iNext*; third, the raw effective species number; fourth, excluding cities with fewer than 10,000 trees; fifth, excluding cities with <50% spatial coverage; sixth, excluding cities with <0.995 sample coverage as calculated by *iNext*. For the fourth, fifth, and sixth models, the independent variable was effective species for a standardized population size of 37,000 trees.”

We further added the following disclaimer text:

“These data havr been collected over many years by urban foresters, citizen scientists, consulting firms, and other interested parties; here, we could not evaluate each city’s accuracy at species identification and location determination. Likewise, we could not fully control for different sampling schemes and sampling efforts (but see ‘Materials and methods’). Future work could deploy tree experts to randomly resample trees in each city and compare the identification to that in our dataset.”

Line 117. Effective species count is for measuring diversity, not biodiversity. Please see the Lou Jost's website as listed in the paper (Line 369-370).

We changed this, and similar instances, to “species diversity” or “diversity”

Line 210-211. What is the relationship between considering hardiness when selecting tree species and tree health metrics? The health metrics used by the authors are rather general, not designed specifically for measuring a tree specie's hardiness.

We rephrased this confusing sentence as follows:

“Urban foresters typically aim to select tree species which will be healthy in their city environment. Our dataset provides standardized metrics of tree health across many cities, allowing analyses of what tree-specific or location-specific factors correlate with health in city trees.”

Line 256-259. The interpretation that heat islands are more common in species-poor areas of Washington, DC is a overstretch. There are so many confounding factors, including anthropogenetic heat sources, urban layout, and the tree cover as mentioned by the authors.

We agree, and have changed this paragraph to read:

“City trees cool urban temperatures (Kong et al., 2014) and clean the air, benefits which are not equitably distributed. For example, Figure 5D shows the location of heat islands, which tend to be more common in low-income (and less well-forested) areas of Washington, DC. The dataset herein could be combined with many physical variables for new analyses of how tree diversity and species compositions relate to temperature, air quality, and more.”

Line 271. The statement should be rephrased. The human can never exercise precise control of urban forests. Many biotic and abiotic factors that we cannot control impact urban forests. The climate factors discussed by the authors are a good example.

We have changed it to read “Urban forests are ecosystems over which humans exercise control, in part by selecting and planting city trees.”

Line 313-356. Tree inventories can be generated using different sampling schemes, e.g., census or probability-based sampling methods. I assumed that the authors had the details in the supplementary files. However, it will help readers if the authors can briefly introduce how these different types of inventories were processed in the main text, especially on these incomplete tree inventories.

See above for details on our substantial new analyses, text, and results.

Line 360. Please explain why the resolution of 2.5 degrees was used. 2.5 degrees roughly equal 250 km. Probably no single US city has such a dimension. WorldClim data are available at resolutions as high as 30 arcseconds (~1 km). It may be more suitable for studies on a city scale.

We did so to save computing time, capture a snapshot of environment in that local region, and ensure that we were covering a city’s entire extent. We confirmed that these average environmental variables at 2.5 degrees resolution are significantly correlated with those at the higher resolution of 0.5 degrees. We added the following sentence:

“We used resolution = 2.5 degrees, and as a sensitivity test we confirmed that these environmental values were significantly correlated with the same values at 0.5 degrees resolution.”

We would be happy to rerun the analyses at 0.5 degrees resolution if you prefer.

In Author response image 3 are some plots demonstrating the correlation between resolution levels:

**Author response image 3. sa2fig3:** 

Line 482. Please explain why use temperature >95o F as the threshold value? The authors used land surface temperature derived from satellite data. It described the thermal patterns of urban surface, not urban heat island.

95 degrees F (35 degrees C) is the physiological upper limit for humans, marking major risk to health and productivity. (https://www.science.org/doi/10.1126/sciadv.aaw1838). A more sophisticated analysis would of course need to be done for a paper focusing on heat risks to humans.

We included the information that we used land surface temperatures in the figure caption so that readers know exactly what they are looking at. Other papers focusing on the phenomenon or urban heat islands also use landsat data (https://doi.org/10.1080/01431169208904271). As it stands, we feel that the caption is accurate, but please let us know if you would suggest wording changes to be more precise.

Reviewer #3 (Recommendations for the authors):Not totally necessary, but may want to consider the using the terms "introduced" vs. "naturally occurring" species rather than “native” vs. “non-native”. There are some arguments from humanities folks about the history of the term native in colonialist narratives. Leaving it as is may distract from the very real biological argument that you are trying to make about the evolutionary history of locally interacting species leading to more interactions being supported by naturally occurring species than those that humans introduced more recently.

Good thinking. We have made this change throughout, clarifying in a few places that we are referring to what is traditionally called “native”.

"…validate these tools in comparison to past methods…"- It is unclear to me from the text how validation was done.

We changed this to read: “compare our results to findings based on past methods”, referring to how we find other axes of interest (e.g., spatial composition and effective species number) which are not captured in traditional tools (such as those that calculate maximum abundance of a single species, genus, and family).

Figure 4- what does the green line in the right panel of D represent- it isn't defined in the caption…

We added the following explanation: “In the right panel, the green line is the same as in the left panel to enable comparisons.”

As also mentioned in the paper- you may want to consider specifying in your captions text etc that it is Shannon's effective species number you are using. Effective species can cover all hill numbers with q as a scaler (i.e. Simpson's effective species number is also an "effective species count")

Good idea– we added this clarification to the text and to the figure caption as follows:

“…effective species counts (a robust measure of diversity defined as the exponent of the Shannon-Weiner index; Equation 1)”.

We use Shannon’s effective species count (Equation 1), a more nuanced metric than abundance-based metrics (see supporting Figure S2).

The excel supplemental tools need directions on the form itself to make them usable

We have added instructions in the first tab on the sheet. Good idea!

Data S2-– Some "most_common_species" are a genus instead (i.e Santa Rosa, Detroit). What is going on.

Some city’s inventories only identified species to the genus level. We decided to include these in the “most common” calculations, because from talking to city foresters, often they are indeed one species. We hope that future work can improve the specificity of the dataset.

– 4 columns are all N/As- why are they included?

We deleted these columns- thank you.

Data S6 tool doesn't work for me. I pick my state and put in species and it doesn't work

We have revised the Excel code to be more robust, and added instructions- does it work now?

Data S3 tool doesn't work either- just get a Div/0 error if you modify the species list.

We have revised the Excel code to be more robust, and added instructions- does it work now?

Figure S2. "Effective species is a more nuanced metric of biodiversity than classic abundance-based measures". Define what classic abundance-based means. I think you mean the % of most common species which is I believe related to the Berger-Parker dominance Index. The confusion lies because for a community ecologist effective species number (i.e. exponent of Shannon's diversity) is a classic index that takes into account relative abundance as well as richness.

Good point- we shouldn’t use the word classic, since it is too general. Also, as you point out, effective species number is classic for many potential readers of this paper! We have changed this to read:

“Effective species is a more nuanced metric of species diversity than the metric of maximum relative abundance of a single species or genus”.

Posting R code to github as well would substantially improve the usability of the code for more sophisticated users and allow potential modifications as datasets are added.

We agree- we are posting the code to github!

Figure 5- "Land surface temperature in July 2018 shows that the highest temperatures, and 290 urban heat islands with temperatures > 95ºF, tend to overlap with less-richly-forested areas."- This figure does not show forest cover and I don't believe the DC data set includes every tree in urban forests so this statement does not seem supported by the analysis as currently completed.

We have changed the caption to read:

“Land surface temperatures in July 2018 are plotted to show the spatial location of the highest temperatures, including urban heat islands with temperatures > 95ºF.”

And we changed the in-text discussion to read:

“For example, Figure 5D shows the location of heat islands in Washington, DC; urban heat islands can be mitigated by planting city trees and increasing canopy cover (Gartland, 2012)”.

[Editors' note: further revisions were suggested prior to acceptance, as described below.]

The manuscript has been substantially improved. We appreciate all the efforts made by the authors. There are some remaining issues that need to be addressed, as outlined below:1. L177: the numbers of effective species counts were inconsistent with those presented in the legend of Figure 2. Please make the correction.

We changed the numbers in Line 177 to match Figure 2, so that it now reads “min=6 to max=93 with a median=26.”

2. Please provide the degrees of freedom (df) for the results of paired t-test in the legends of Figure 2, Figure 2—figure supplement 1, Figure 4.

We added the following degrees of freedom information.

Figure 2 B legend: degrees of freedom = 10.Figure 2—figure supplement 1: degrees of freedom = 10

We added a note to the figure that explains why it looks like there are only 10 points:

“Note: the plot includes 11 cities, but Denver, CO and Aurora, CO overlap at this level of resolution.”

Figure 4 C: degrees of freedom = 1,952Figure 4 D: degrees of freedom = 1,952

3. L227-228: please clarify environmental PC1 and cite the PCA results (supplementary file 5) in the main text before introducing this result.

L196, we added to the main text: “We summarized the climate of each city with a principal components analysis (PCA) of 19 bioclimatic variables from the WorldClim database (Supplementary File 5).”

4. Figure 3: there is one city having a very large 95%CI, overlapping both the lines of zero and 100%. This case is very strange. How could this happen? How is possible that an observed effective species per cluster = 0?

Thank you for noting this. That city, Greensboro, had a small population size and only 10 clusters. That confidence interval is a mathematical calculation based on the standard error and degrees of freedom (see code copied below from Source_Code_File_2). The confidence interval just mathematically represents error and is agnostic to the fact that in this case, the data we are dealing with is ratios.

Therefore, we agree with you that we need to think again about this result, and we decided on the following interpretation: in this case, that large confidence interval indicates that clustering results for Greensboro do not represent biological reality; therefore, we have cut this city from the analysis and added a note to the figure caption. The note to the Figure 3 caption reads:

“We excluded one city Greensboro, from the analysis due to insufficient sample size (10 clusters).”

Relatedly, we realized that we have 48 total cities for which we conducted a clustering analysis, so we changed “44 of 46” to “47 of 48” throughout as well as changed “trees significantly cluster by species in 96% of cities” to “98% of cities”.

Figure 3 captionIn-text, L256Abstract, L51

Code to calculate confidence interval:

n <- length(ratio)median_value <- median(ratio)standard_deviation <- sd(ratio)standard_error <- standard_deviation / sqrt(n)α = 0.05degrees_of_freedom = n – 1t_score = qt(p=α/2, df=degrees_of_freedom,lower.tail=F)margin_error <- t_score * standard_errorlower_CI <- median_value – margin_errorupper_CI <- median_value + margin_error

5. L423: should be "what are our null expectations".

Change made.

6. L1065: please clarify whether the effective species numbers were raw statistics or based on standardized population size.

Thank you for catching this. We changed this to be effective species numbers based on standardized population size and replotted (the results did not change).

We edited the figure caption as follows:

In the Figure 4—figure supplement 1 caption, we deleted the text “; in both cases, we calculated effective species numbers.”

We instead added the text: “To allow for comparison across cities with different sizes and sampling efforts, we plot the calculated effective species number for the rounded median population of all trees (for all species, 37,000 trees; x-axis) and native species only (10,000 trees; y-axis) – calculated using rarefaction and extrapolation in R package *iNext*.”

7. L1107: should be in "Figure 4A".

Change made.